# DREAM: Dual Structured Exploration with Mixup for Open-set Graph Domain Adaption

**Nan Yin[1]\*, Mengzhu Wang[2]\*, Li Shen[3], Zhenghan Chen[4], Huan Xiong[1], Bin Gu[1], Xiao Luo[5]†**
[1] Mohamed bin Zayed University of Artificial Intelligence   [2] Hebei University of Technology
[3] JD Explore Academy   [4] Peking University   [5] University of California, Los Angeles
{yinnan8911,dreamkily,mathshenli,pandaarych}@gmail.com
{huan.xiong,bin.gu}@mbzuai.ac.ae, xiaoluo@cs.ucla.edu

## Abstract

Recently, numerous graph neural network methods have been developed to tackle domain shifts in graph data. However, these methods presuppose that unlabeled target graphs belong to categories previously seen in the source domain. This assumption could not hold true for in-the-wild target graphs. In this paper, we delve deeper to explore a more realistic problem *open-set graph domain adaptation*. Our objective is to not only identify target graphs from new categories but also accurately classify remaining target graphs into their respective categories under domain shift and label scarcity. To solve this challenging problem, we introduce a new method named Dual Structured Exploration with Mixup (DREAM). DREAM incorporates a graph-level representation learning branch as well as a subgraph-enhanced branch, which jointly explores graph topological structures from both global and local viewpoints. To maximize the use of unlabeled target graphs, we train these two branches simultaneously using posterior regularization to enhance their inter-module consistency. To accommodate the open-set setting, we amalgamate dissimilar samples to generate virtual unknown samples belonging to novel classes. Moreover, to alleviate domain shift, we establish a k nearest neighbor-based graph-of-graphs and blend multiple neighbors of each sample to produce cross-domain virtual samples for inter-domain consistency learning. Extensive experiments validate the effectiveness of the proposed DREAM in comparison to various state-of-the-art approaches in different settings.

## 1 Introduction

The ubiquity of graph-structured data on the Internet has captivated the interest of the graph machine learning community (Yoo et al., 2022; Yang et al., 2022; Zhang et al., 2021; Feng et al., 2022a). Cutting-edge graph neural network (GNN) approaches (Cui et al., 2022; Liu et al., 2022; Feng et al., 2022b) have recently demonstrated exceptional capabilities in graph representation learning. Typically, these methods employ the message-passing mechanism to generate effective node-level representations. Then, a range of graph pooling procedures (Yin et al., 2023b; Ying et al., 2018; Lee et al., 2019; Bianchi et al., 2020; Wang et al., 2020) has been devised, offering graph-level representations embedded with topological semantics for numerous downstream tasks including graph property prediction (Lu et al., 2019; Wieder et al., 2020; Chen et al., 2021; Li et al., 2022b).

Despite their promising performance, the majority of these approaches rely on supervised learning, requiring a substantial volume of labeled data. To mitigate the burden of data annotation, a variety of graph domain adaptation approaches have been devised (Yin et al., 2022; Yehudai et al., 2021). These methods enhance the model by integrating labeled graphs from a distinct yet related source domain. However, these methods presuppose that source and target graphs share the same label space, which often does not hold true in practical scenarios (Scheirer et al., 2012; 2014; Rizve et al., 2022). Consequently, trustworthy GNN systems should be endowed with the ability to infer what they have never encountered before. In academic literature, out-of-distribution (OOD) detection

---

*Equal contribution.
†Corresponding author.

has been extensively studied in various safety-critical applications (Amodei et al., 2016) including medical diagnosis (Kukar, 2003), autonomous driving (Dai & Van Gool, 2018), and among others. Although several intriguing strategies have been proposed for OOD detection in visual and text domains (Bitterwolf et al., 2022; Panareda Busto & Gall, 2017; Tan et al., 2019), this problem remains underexplored in graph-structured data. Towards this end, this paper investigates the problem of *open-set graph domain adaptation*, which involves identifying OOD target graphs from unknown classes and classifying the remaining graphs into their corresponding classes.

In fact, the problem is quite different owing to the reasons as follows: (1) *Semantic Drift.* Traditional graph domain adaptation techniques (Yin et al., 2022; Yehudai et al., 2021; Wu et al., 2020a) often utilize pseudo-labeling to extract knowledge from unlabeled target graphs. However, substantial domain shift and label shift can generate biased pseudo-labels for target graphs, leading to an accumulation of errors during subsequent optimization. (2) *Inadequate Supervision.* Extensively labeled in-distributions (ID) samples are accessible in the majority of out-of-detection methods (Hein et al., 2019; Hendrycks et al., 2018). However, our problem could encounter both data scarcity and label scarcity on the target domain as graph data is more difficult to acquire, which would increase the difficulty of identifying target samples from novel classes. As a consequence, a powerful framework that can extract sufficient topological information from unlabeled graph samples to provide additional high-quality supervision is highly desired.

In this paper, we propose a novel framework named Dual Structured Exploration with Mixup (DREAM) for open-set graph domain adaption. The key idea of our DREAM is to intensively mine graph topological structures using complementary branches (i.e., the graph-level representation branch and subgraph-enhance branch), which are then incorporated into a trustworthy and domain-invariant framework. On the one hand, our graph-level representation branch employs a message-passing mechanism to encode topological knowledge into node representations, subsequently employing an attention mechanism for their aggregation from diverse perspectives. On the other hand, our subgraph-enhanced branch composites each graph into several subgraphs using graph clustering. Then, hierarchical GNNs are utilized to extract localized functional components and their corresponding interactions. We amalgamate the strengths of both branches by employing posterior regularization, which enhances the coherence between the predictions from the two branches and thus mitigates biased and overconfident pseudo-labels. Furthermore, to accommodate the open-set scenario, we blend dissimilar source samples in the latent space, generating virtual OOD samples for additional supervision. To further mitigate domain shift, we construct a $k$-nearest neighbor-based graph-of-graph, where nodes represent graph samples. Multiple neighbors of each graph sample are then combined to generate a cross-domain virtual sample. We strongly encourage inter-domain consistency in the predictions between each sample and its corresponding cross-domain counterpart. Our proposed DREAM has demonstrated remarkable effectiveness when compared to state-of-the-art approaches in various challenging scenarios. Moreover, we conduct extensive ablation studies and visualization to validate the superiority of our DREAM.

In summary, our paper makes the following contributions: (1) *Problem Formulation*: We introduce a novel problem of open-set graph domain adaptation, which accommodates unlabeled in-the-wild target graphs from unseen classes. (2) *Methodology*: We introduce an approach called DREAM that employs two branches to investigate structural semantics and integrates them into a trustworthy and domain-invariant framework. (3) *Experiments.* Comprehensive experiments verify the effectiveness of our proposed DREAM by comparing state-of-the-art approaches.

## 2 RELATED WORK

**Graph Neural Networks (GNNs).** GNNs (Kipf & Welling, 2017; Bodnar et al., 2021; Qu et al., 2019; He et al., 2022; Baek et al., 2021) have proven to be remarkably effective in handling graph-based machine learning tasks including node classification (Cui et al., 2022; Liu et al., 2022; Feng et al., 2022b), link prediction (Nguyen et al., 2022; Zhao et al., 2022; Zhang et al., 2019), and graph classification (Fan et al., 2019; Song et al., 2016; Liao et al., 2021; Wu et al., 2020c). Recently, graph neural networks have also been studied in different OOD settings. Semi-supervised open-world graph classification involves partial unlabeled graphs belonging to unknown classes (Luo et al., 2023). Graph OOD detection aims to detect OOD graph samples without using ground-truth labels (Liu et al., 2023). Node-level open-world graph learning aims to find OOD nodes on a single

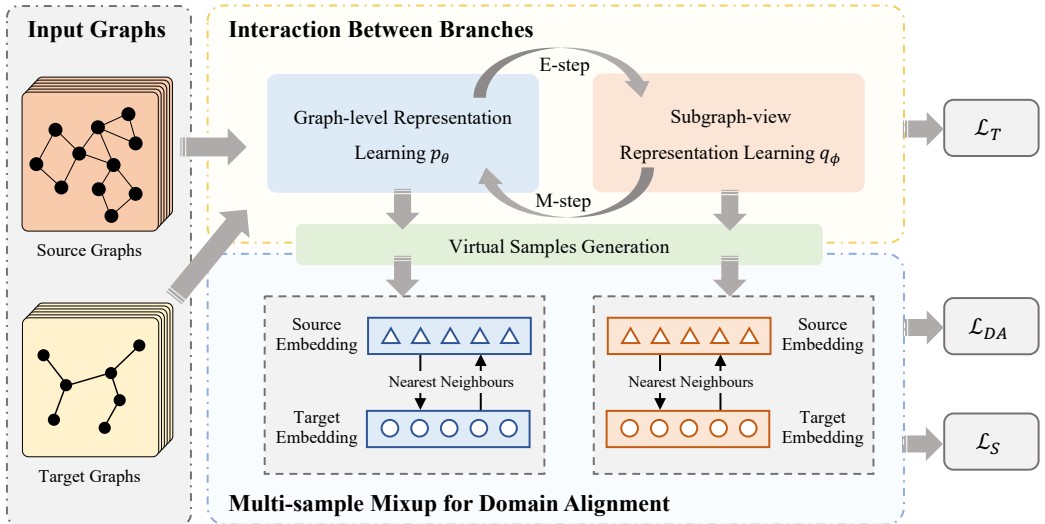

Figure 1: An overview of the proposed DREAM. We feed both source and target graphs into both the graph-level representation branch and subgraph-view representation learning branch. We utilize a unified EM-based framework for interaction between two branches. Dissimilar source samples are mixed to simulate samples from novel classes. Moreover, we utilize multi-sample mixup to generate cross-domain virtual samples for domain alignment.

graph (Wu et al., 2020b; Hoffmann et al., 2023). Compared with these problem settings, our problem is more challenging, which not only detects OOD graph samples, but also overcomes distribution shifts (Ju et al., 2024; Gui et al., 2022; Buffelli et al., 2022) across source and target domains.

**Graph Domain Adaption.** Domain adaptation is a vital problem in the field of machine learning (Huang et al., 2022b; Zhao et al., 2021; Pang et al., 2023), which strives to transfer knowledge from a label-rich source domain to a target domain suffering from label scarcity. This problem has also been explored in graph domains, and various graph transfer learning approaches are developed that incorporate adversarial learning (Dai et al., 2022; Wu et al., 2020a) and pseudo-labels (Yin et al., 2022; Yehudai et al., 2021) into GNNs. However, these approaches are under the assumption that the label spaces of the source graph and target graphs are identical, which often does not hold true in practical scenarios (Panareda Busto & Gall, 2017; Chen et al., 2022). To address this, our study introduces an under-explored problem of open-set graph domain adaptation and formulates a novel approach DREAM to tackle this problem.

**Open-set Recognition.** The objective of open-set recognition is to exclude instances that belong to novel classes absent in the training data (Geng et al., 2020). Current efforts can be categorized into generative and discriminative approaches. Generative methods use conditional auto-encoders (Oza & Patel, 2019) and data augmentation (Ditria et al., 2020) to simulate the distribution of novel classes, while discriminative models (Scheirer et al., 2012; 2014; Rizve et al., 2022) build distinct classifiers to detect outliers. Open-set recognition can also be combined domain adaptation settings (You et al., 2019; Chen et al., 2022). Notwithstanding its significance, this problem has primarily been studied in the context of Euclidean data, and remains unexplored in graph-structured data. To the best of our knowledge, we are the first to study open-set graph domain adaptation.

## 3 THE PROPOSED DREAM

**Problem Setup.** A graph is depicted as $G = (\mathcal{V}, \mathcal{E})$ with the node set $\mathcal{V}$ and the edge set $\mathcal{E}$. The node attribute matrix is represented by $\boldsymbol{X} \in \mathbb{R}^{|\mathcal{V}| \times F}$. We are given $n_s$ labeled graphs from a source domain $\mathcal{D}^s = \{(G_i^s, y_i^s)\}_{i=1}^{n_s}$ and $n_t$ unlabeled graphs $\mathcal{D}^t = \{G_j^t\}_{j=1}^{n_t}$. The source and target label spaces are $\mathcal{Y}^s$ and $\mathcal{Y}^t$, respectively. We assume $\mathcal{Y}^s \subset \mathcal{Y}^t$ instead of $\mathcal{Y}^s = \mathcal{Y}^t$ in classic domain adaptation settings. Our aim is to not only classify known target graphs into their corresponding classes, but also identify target graphs from novel classes $\mathcal{Y}^t/\mathcal{Y}^s$.

## 3.1 FRAMEWORK OVERVIEW

This work proposes a new method named DREAM for open-set graph domain adaption. Recognizing that subgraphs can provide additional topological information for graph classification under domain shift (Alsentzer et al., 2020), our DREAM introduces two complementary branches, i.e., a graph representation learning branch and a subgraph-enhanced branch to learn graph representations. We then integrate these two branches into a unified EM-based framework, which enhances their inter-branch consistency on target graphs to reduce biased and overconfident pseudo-labels. Additionally, we blend dissimilar source graphs in the latent space to simulate samples from novel classes, thereby enhancing the capacity to identify OOD samples. To further alleviate the impact of domain shift, we construct a k nearest neighbor-based graph-of-graph and employ multi-sample mixup to transform graphs into virtual cross-domain samples for inter-domain consistency learning. A detailed illustration of the proposed DREAM can be found in Figure 1.

## 3.2 GRAPH-LEVEL REPRESENTATION LEARNING FOR OPEN-SET CLASSIFICATION

The preliminary point of graph classification is to map each graph sample into a compact embedding. Previous methods usually utilize GNNs and graph pooling operators (Kipf & Welling, 2017; Veličković et al., 2018; Xu et al., 2019; Baek et al., 2021; Lee et al., 2019) for graph classification. To involve more multi-view semantics, after generating node representations using GNNs, we employ the attention mechanism (Veličković et al., 2018), which summarizes them into super-node representations from different perspectives, ultimately producing a graph-level representation. Subsequently, to detect OOD samples (Chen et al., 2022; Saito et al., 2020; Li et al., 2023) in the graph domain, we introduce an open-set classifier for identifying novel classes in our problem and then generate virtual samples using manifold mixup techniques (Verma et al., 2019) for extra supervision.

In particular, given a graph sample $G = (\mathcal{V}, \mathcal{E})$, $\boldsymbol{h}_i^{(l)}$ denotes the representation of node $i \in V$ at layer $l$. We first introduce a GNN $\Phi^V(\cdot, \cdot)$ to update node representations:

$$\boldsymbol{n}_i^{(l)} = \text{AGGREGATE}^{(l)} \left( \left\{ \boldsymbol{h}_j^{(l-1)} : j \in \mathcal{N}(i) \right\} \right), \boldsymbol{h}_i^{(l)} = \text{COMBINE}^{(l)} \left( \boldsymbol{h}_i^{(l-1)}, \boldsymbol{n}_i^{(l)} \right), \quad (1)$$

where $\mathcal{N}(i)$ denotes the neighbours of $i$. $\text{AGGREGATE}^{(l)}(\cdot)$ and $\text{COMBINE}^{(l)}(\cdot)$ denote aggregation and combination operators at layer $l$, respectively. After stacking $L$ layers, we can generate final node representation matrix $\boldsymbol{H} = \Phi^V(G, \boldsymbol{X}) = [\boldsymbol{h}_1^L, \cdots, \boldsymbol{h}_{|V|}^L]$. Instead of directly summarizing these node representations, we adopt the attention mechanism to generate super-nodes that reflect different aspects of graphs. Here, we introduce a query matrix $\mathcal{Q} \in \mathbb{R}^{T \times d}$ and then introduce a different GNN $\Phi^o(\cdot, \cdot)$ to generate $T$ views of graphs. In formulation,

$$\boldsymbol{\Gamma} = \text{softmax} \left( \frac{\mathcal{Q} \cdot \mathcal{K}^\top}{\sqrt{d}} \right) \cdot \boldsymbol{H}, \quad \mathcal{K} = \Phi^o(G, \boldsymbol{X}), \quad (2)$$

where $\boldsymbol{\Gamma} = [\boldsymbol{\gamma}_1, \cdots, \boldsymbol{\gamma}_T] \in \mathbb{R}^{T \times d}$ collects $T$ super-node representations of $G$. Finally, we concatenate all these representations into matrix $\boldsymbol{\Lambda} \in \mathbb{R}^{Td}$, and then utilize an MLP to summarize the information in $\boldsymbol{\Lambda} \in \mathbb{R}^{Td}$ into a graph-level representation $\boldsymbol{h}^g = \mathcal{F}^g(G)$ where $F^g$ denotes the whole graph-level representation learning branch.

**Open-set Classifier.** To connect graphs into their corresponding classes, we introduce a classifier $\psi^g(\cdot)$, which maps graph representations into label distributions, i.e., $p_\theta(y|G) = \psi^g(\boldsymbol{h}^g) = \psi^g(\mathcal{F}^g(G))$ with parameters $\theta$. Further, we extend the closed-world classifier by introducing an extra dimension, which outputs the probability of graphs belonging to novel classes. Since we do not get access to labeled from the unseen classes, we turn to manifold mixup techniques (Verma et al., 2019). Here, we choose two graph samples from different classes and then leverage linear interpolation to generate virtual samples in the hidden space. Given $y_i^s \neq y_j^s$, we have:

$$\bar{\boldsymbol{h}} = \lambda \mathcal{F}^g(G_i^s) + (1 - \lambda) \mathcal{F}^g(G_j^s), \quad (3)$$

where $\lambda$ is sampled from $(0, 1)$. After generating a set of virtual samples $\mathcal{D}^v$ with their labels $C + 1$, we will train the whole neural network by minimizing the following objective:

$$\mathcal{L}_S = -\mathbb{E}_{(G,y) \in \mathcal{D}^s} [\log p_\theta(y|G)] - \mathbb{E}_{(\bar{\boldsymbol{h}},y) \in \mathcal{D}^v} [\log \phi_\theta^g(\bar{\boldsymbol{h}})[C + 1]]. \quad (4)$$

### 3.3 SUBGRAPH-VIEW REPRESENTATION LEARNING FOR SEMANTIC ENHANCEMENT

Due to potential domain shift and label scarcity on the target domain, our graph-level representations could be biased for target data. Note that subgraph and patching strategies have been popular in graph representation learning (Han et al., 2022; He et al., 2022; Luo et al., 2022), which are underexplored in graph domain adaptation. Towards this end, we introduce a subgraph-level representation learning branch to provide a complementary view, which can focus on key local functional parts in graphs and generate graph representation using a hierarchical GNN architecture.

In detail, we leverage the graph clustering algorithm (i.e., Metis (Karypis & Kumar, 1998)) to generate $R$ subgraphs as follows:

$$[\tilde{G}^1, \cdots, \tilde{G}^{l_R}] = \left[\left\{V^1, E^1\right\}, \cdots, \left\{V^R, E^R\right\}\right],\tag{5}$$

where each $E^r$ is comprised of edges between nodes in $V^r$. Then, we adopt a GNN $\Phi^S(\cdot)$ to generate all the subgraph representations using $\tilde{\boldsymbol{h}}_r = \text{SUM}(\Phi^S(\tilde{G}^{l_r}, \tilde{\boldsymbol{X}}^{l_r}))$ where $\text{SUM}(\cdot)$ summarizes updated node representations. Then, we calculate a new adjacency matrix to model the interaction between subgraphs. In particular, we have:

$$\tilde{\boldsymbol{A}}_{r_1 r_2} = \sum_{i \in \tilde{G}^{r_1}} \sum_{j \in \tilde{G}^{r_2}} \boldsymbol{A}_{ij},\tag{6}$$

where $\boldsymbol{A}_{ij}$ is the weight between $i$ and $j$ in the original graph. Given these new graphs with the stacked subgraph representation matrix $\tilde{\boldsymbol{H}} \in \mathbb{R}^{R \times d}$, we adopt a GNN $\Phi^F$ to model the interaction between subgraphs and summarize them into a graph-level representation, i.e., $\boldsymbol{h}^s = \text{SUM}(\Phi^F(\tilde{G}, \tilde{\boldsymbol{H}}))$. Similarly, we can utilize a different classifier $\psi^s(\cdot)$ to generate $q_\phi(y|G) = \psi^s(\boldsymbol{h}^s) = \psi^s(\mathcal{F}^s(G))$ where $\mathcal{F}^s$ summarizes the subgraph-enhanced branch with parameters $\phi$.

### 3.4 INTERACTION BETWEEN TWO BRANCHES

We have built a graph-level representation learning branch and a subgraph-enhanced branch. In this part, we combine the advantages of these two branches by posterior regularization (Ganchev et al., 2010; Qu et al., 2019; Lin et al., 2019) to enhance the consistency between the predictions from the two branches.

In detail, we propose to maximize the log-likelihood of graph representation learning, i.e., $\mathbb{E}_{G \in \mathcal{D}^t}[\log p_\theta(G)]$. Then, we have the following lemma with the posterior distribution $q_\phi(y|G)$:

**Lemma 3.1.** $\mathbb{E}_{G \in \mathcal{D}^t}[\log p_\theta(G)] \geq \mathbb{E}_{G \in \mathcal{D}^t, y \sim q_\phi(y|G)}[\log \frac{p_\theta(G,y)}{q_\phi(y|G)}]$ *where the equality holds when* $p_\theta(y|G) = q_\phi(y|G)$.

The proof is given in Appendix A. Therefore, we minimize the KL divergence between the prediction from two branches on the target domain:

$$\mathbb{E}_{G \in \mathcal{D}^t} KL(q_\phi(y|G)||p_\theta(y|G)).\tag{7}$$

To minimize the inconsistency, we adopt an EM-style optimization manner. In particular, we first update the graph-level representation learning branch, and then update the subgraph-enhanced branch.

**E-step.** We first fix $\theta$ and update the posterior distribution, i.e., the subgraph-enhanced branch. Here, we adopt the reserved KL divergence to simplify the calculation, i.e., $\min KL(p_\theta(y|G)|q_\phi(y|G))$. Note that $\mathbb{E}_{G \in \mathcal{D}^t, y \sim q_\phi(y|G)}[\partial \log q_\phi(y|G)/\partial \theta] = 0$, we have:

$$\partial \mathbb{E}_{G \in \mathcal{D}^t} KL(p_\theta(y|G)|q_\phi(y|G))/\partial \phi = \partial \mathbb{E}_{G \in \mathcal{D}^t, y \sim (p_\theta(y|G)+q_\phi(y|G))} \left[-\log q_\phi(y|G)\right]/\partial \phi.\tag{8}$$

From Eqn. 8, we generate the pseudo-labels of target graphs based on the prediction from both two branches, which can reduce the overconfident pseudo-labels through their collaboration.

**M-step.** Then, we update the graph representation learning branch with $\phi$ fixed. Similarly, we have:

$$\partial \mathbb{E}_{G \in \mathcal{D}^t} KL(q_\phi(y|G)|p_\theta(y|G))/\partial \theta = \partial \mathbb{E}_{G \in \mathcal{D}^t, y \sim (p_\theta(y|G)+q_\phi(y|G))} \left[-\log p_\theta(y|G)\right]/\partial \theta.\tag{9}$$

In summary, we generate confident pseudo-labels for target graphs using both two branches, which takes the intersection of confident pseudo-labels from two branches Then, they are utilized to optimize our two branches in an alternative manner with:

$$\mathcal{L}_T = \mathbb{E}_{G \in \mathcal{D}^t, y \sim (p_\theta(y|G)+q_\phi(y|G))} \left[-\log p_\theta(y|G) - \log q_\phi(y|G)\right].\tag{10}$$

### 3.5 MULTI-SAMPLE MIXUP FOR DOMAIN ALIGNMENT

However, the challenge of serious domain shift in our problem still exists, which could potentially introduce biased pseudo-labels. To tackle this issue, we propose a multi-sample mixup strategy (Zhang et al., 2022; Verma et al., 2019), which combines multiple cross-domain neighboring samples for virtual samples. With the assumption of similar semantics for close graph representations (Long et al., 2018; Luo et al., 2021; 2023; Zhao et al., 2021), we aim to maximize the consistency of predictions between original and virtual samples for domain alignment.

In detail, we first identify cross-domain neighbors in the whole dataset. Here we take the first branch as an example and omit the superscript of graph representations. Since there would be target graphs from novel classes, we utilize a $k$ mutual nearest neighbor graph-of-graph $\mathcal{G}$ where each node represents a graph sample in the dataset. To save the computational cost, we introduce a memory bank $\mathcal{M}$ with size $|\mathcal{M}|$, which is updated in a first-in-first-out fashion. The adjacency matrix $\mathcal{A}$ is:

$$\mathcal{A}_{ij} = \left\{ \begin{array}{ll} 1 & G_i^s \in \mathcal{T}_k(G_j^t) \subset \mathcal{M} \vee G_j^t \in \mathcal{T}_k(G_i^s) \subset \mathcal{M} \\ 0 & \text{otherwise,} \end{array} \right. \tag{11}$$

where $\mathcal{T}_k(\cdot)$ collecting $k$-nearest neighbours in the hidden space. Then, for each source graph sample $G_i^s$, we take the combination of all neighbours in $\mathcal{G}$:

$$\boldsymbol{z}_i^s = \sum_{\mathcal{A}_{ij}=1} \lambda_j^i \boldsymbol{h}_j^t, \tag{12}$$

where $\lambda_j^i = s(G_i^s, G_j^t) / \sum_{j'} s(G_i^s, G_{j'}^t)$ and $s(G_i^s, G_j^t)$ denotes the cosine similarity between two graphs in the hidden space. Similarly, we can generate cross-domain virtual features for each target graph. Then, we maximize the consistency between the original graphs and their corresponding cross-domain virtual ones. In particular, we have:

$$\mathcal{L}_{DA} = \mathbb{E}_{(G,y) \in \mathcal{D}^s \cup \mathcal{D}^t} KL(p(y|G) || p(y|\boldsymbol{z})), \tag{13}$$

where $\boldsymbol{z}$ denotes the cross-domain features generated by $G$.

**Framework Summarization.** Finally, the overall training objective can be summarised as:

$$\mathcal{L} = \mathcal{L}_S + \alpha \mathcal{L}_T + \beta \mathcal{L}_{DA}, \tag{14}$$

where $\alpha$ and $\beta$ are hyper-parameters. We utilize mini-batch stochastic gradient descent (SGD) to update the whole DREAM. The detailed algorithm is shown in Algorithm C.

## 4 EXPERIMENT

### 4.1 EXPERIMENTAL SETTINGS

**Evaluation Datasets.** To validate the effectiveness of DREAM, we perform extensive experiments on six widely used graph classification datasets from TUDataset and BenchmarkDataset, including MSRC_21 (Neumann et al., 2016), Letter-high (Riesen et al., 2008), COIL-DEL (Nene et al., 1996; Riesen et al., 2008), COIL-RAG (Riesen et al., 2008; Nene et al., 1996), MNIST (Dwivedi et al., 2020), and CIFAR10 (Dwivedi et al., 2020). The specific statistics and details introduction of experimental datasets are presented in Appendix D.

**Evaluation Protocols.** Under the open-set domain adaptation settings, we first divide the classes into known and unknown classes which are shown in Appendix D, and then we divide datasets into different domains (i.e., source and target domains) by the density of graphs (i.e., the ratio between graph nodes and edges) indicated by $P1$ and $P2$ (Li et al., 2022a) and the target domain also consists of unknown graphs. Specifically, in the open-set setting, we treat all the unknown classes as a unified class, and when the prediction of samples belonging to known classes with the correct labels or novel classes with the unified class, we declare they are correct.

**Compared Methods.** We compare our proposed DREAM with a wide range of state-of-the-art approaches, including seven graph classification methods (i.e., GMT (Baek et al., 2021), Graph-Mix (He et al., 2022), GCN (Kipf & Welling, 2017), GraphSage (Hamilton et al., 2017), GIN (Xu et al., 2019), SAG Pooling (Lee et al., 2019) and GraphCL (You et al., 2020)), as well as three of

Table 1: Classification accuracy comparisons on four benchmark graph classification datasets (I). The best results are in boldface. $P1$ and $P2$ denote different domains in the datasets.

| Methods | MSRC_21 | | Letter-high | | COIL-DEL | | COIL-RAG | |
|---|---|---|---|---|---|---|---|---|
| | $P1 \to P2$ | $P2 \to P1$ | $P1 \to P2$ | $P2 \to P1$ | $P1 \to P2$ | $P2 \to P1$ | $P1 \to P2$ | $P2 \to P1$ |
| GMT | 63.3±3.7 | 62.3±1.5 | 45.8±2.8 | 40.7±1.4 | 34.6±3.1 | 27.9±1.7 | 57.4±1.1 | 60.6±1.3 |
| GraphMix | 64.5±2.8 | 65.7±2.4 | 46.0±2.3 | 45.6±1.1 | 36.6±2.6 | 35.8±1.2 | 49.5±1.7 | 50.6±2.1 |
| GCN | 63.5±3.4 | 63.6±2.8 | 40.2±4.1 | 38.6±1.5 | 20.6±3.7 | 20.3±2.2 | 58.3±1.7 | 56.8±0.8 |
| GraphSage | 60.5±2.3 | 54.1±3.3 | 50.0±3.6 | 45.9±2.0 | 24.8±4.1 | 14.6±2.5 | 58.5±2.2 | 54.7±1.4 |
| GIN | 66.3±3.2 | 63.3±2.7 | 47.3±2.2 | 46.2±1.0 | 30.6±2.3 | 13.9±1.3 | 58.0±0.7 | 52.9±0.7 |
| SAG Pooling | 61.7±4.4 | 56.9±4.2 | 44.9±3.1 | 40.4±0.7 | 24.1±2.9 | 11.4±1.7 | 58.5±2.3 | 54.8±1.5 |
| GraphCL | 67.2±3.3 | 67.0±3.1 | 47.3±2.4 | 46.2±0.8 | 30.6±2.1 | 23.9±1.1 | 58.6±1.8 | 56.2±2.3 |
| CSSR | 73.8±3.3 | **75.8±2.7** | 49.8±3.3 | 42.5±2.2 | 37.0±3.4 | 36.2±0.8 | 54.9±2.1 | 58.0±2.6 |
| DIAS | 70.4±2.9 | 68.8±2.1 | 52.4±4.2 | 52.6±1.3 | 30.8±3.3 | 21.2±1.2 | 57.3±2.4 | 55.7±2.5 |
| OSR | 58.1±4.1 | 61.5±3.4 | 48.1±3.8 | 42.7±1.4 | 33.2±2.0 | 36.4±0.9 | 54.7±4.2 | 51.1±1.2 |
| DREAM | **74.3±5.4** | 75.2±3.7 | **58.7±3.5** | **53.3±0.8** | **44.0±2.6** | **40.2±0.5** | **65.4±1.7** | **62.5±1.9** |
| Improvement | 0.7% | -0.8% | 12.0% | 1.3% | 18.9% | 10.4% | 11.6% | 3.1% |

the most advanced open-set recognition methods (i.e., CSSR (Huang et al., 2022a), DIAS (Moon et al., 2022) and OSR (Vaze et al., 2022)). More details about the compared baselines can be found Appendix E.1. We also compare our proposed DREAM with RIGNN (Luo et al., 2023), OpenWGL (Wu et al., 2020b), OpenWRF (Hoffmann et al., 2023), DEAL (Yin et al., 2022) and CoCo (Yin et al., 2023a) in Appendix F.2.

**Implementation Details.** As for the network architecture, we first pre-train the dual branches (i.e., graph-level and subgraph-enhanced branches) on the labeled domain, followed by training the network using our proposed framework. To avoid memory overflow, we utilize the memory bank mechanism to store the previous embeddings and employ it to train DREAM on these dataset. For the three open-set recognition methods, we use the GIN model as the backbone encoder to obtain the graph-level features. The dropout ratio is set to 0.5. During training, the hidden dimension of all the methods is set to 128, and we use the Adam (Kingma & Ba, 2014) optimizer and set the batch size to 64 as default. The total number of training epochs is 200, and the learning rate is set to 0.001. The hyper-parameters $\alpha$ and $\beta$ are set to 1.0, 1.0, respecitvely.

## 4.2 EMPIRICAL PERFORMANCE

**Overall Comparison.** The open-set domain adaptation accuracy on different datasets compared with baselines is presented in Table 1 and Table 2. From the results, we observe that the proposed DREAM achieves better performance in both $P1 \to P2$ and $P2 \to P1$ scenarios in most cases, which demonstrates the superiority of DREAM. We attribute the reason to two aspects: the enhancement of interaction between two branches and the Mixup strategies that help align different domains. On the one hand, the interaction between two branches helps the model to learn a consistent representation, which fits the framework of EM algorithm (Moon, 1996). On the other hand, the multi-sample mixup mod-

Table 2: Classification accuracy comparisons on two graph classification datasets (II). The best results are in boldface. $P1$ and $P2$ denote different domains in the datasets.

| Methods | MNIST | | CIFAR10 | |
|---|---|---|---|---|
| | $P1 \to P2$ | $P2 \to P1$ | $P1 \to P2$ | $P2 \to P1$ |
| GMT | 64.2±3.4 | 63.5±2.3 | 37.7±2.8 | 37.2±1.3 |
| GraphMix | 65.5±4.1 | 65.7±1.9 | 42.8±3.1 | 43.2±1.4 |
| GCN | 63.4±2.5 | 62.2±2.7 | 26.3±2.1 | 25.8±1.7 |
| GraphSage | 57.0±3.7 | 63.6±3.0 | 27.9±2.7 | 29.4±1.5 |
| GIN | 63.5±2.7 | 62.9±2.4 | 21.7±3.4 | 27.8±0.8 |
| SAG Pooling | 64.2±3.1 | 63.6±1.9 | 28.7±4.1 | 34.0±0.8 |
| GraphCL | 62.8±2.3 | 62.9±1.4 | 27.8±2.8 | 30.9±0.7 |
| CSSR | 70.8±3.0 | 68.2±2.2 | 45.3±2.1 | 44.7±1.1 |
| DIAS | 72.6±2.7 | 70.8±2.7 | 46.3±1.9 | 45.5±0.9 |
| OSR | 74.7±4.2 | 70.2±3.1 | 36.8±2.7 | 32.4±1.6 |
| DREAM | **84.1±2.8** | **81.8±1.5** | **48.3±2.3** | **47.0±0.4** |
| Improvement | 12.6% | 15.5% | 4.3% | 3.3% |

ule generates cross-domain counterparts by linearly combining multiple samples, further reducing domain discrepancy. Moreover, with the enhancement brought by combining dissimilar source samples, DREAM can better classify graph samples in unknown classes. The existing graph classification methods perform worse than DREAM in the open-set task. This is because these methods are

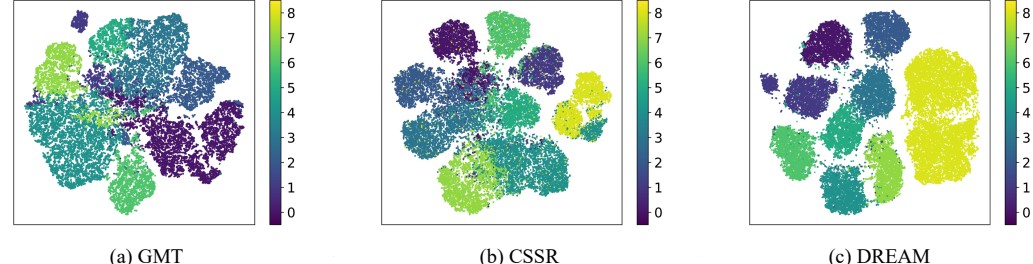

|        (a) GMT        |        (b) CSSR        |        (c) DREAM        |

Figure 2: The T-SNE visualization of latent representation with GMT, CSSR, and DREAM on MNIST dataset. Each point represents a unit's latent representation, and the colors indicate the corresponding labels.

designed for closed-set tasks without the consideration of unknown classes and neglect the semantic shift. Additionally, the results of DREAM outperform the aforementioned open-set recognition methods in most cases. The potential reason is that traditional open-set recognition methods usually use prototype learning to separate the unknown class from all seen classes, which would suffer from the semantic shift and label scarcity in our scenarios.

**Visualization.** For the purpose of achieving an intuitive effect, we demonstrate the visualization of the graph representations for different methods in Figure 2. We visualize the MNIST dataset and compare it with two models (GMT and CSSR). From the results, we can find that the visualization results of the closed-set graph model cannot have an evident conclusion since a wide range of similar graphs are not clustered with huge overlap of points from different classes. This is because the closed-set model cannot effectively separate the unknown samples from the known samples. Compared with GMT, CSSR demonstrates a more structured visualization where we can tell the relationship across different classes. Nevertheless, the boundaries of different classes are still blurry, indicating that although open-set recognition methods can effectively recognize unknown classes, they cannot tackle domain shift and label scarcity in graph data effectively. From the visualization of our proposed DREAM, the graph representations from the same classes are close in comparison to these baselines. In addition, we can observe that the boundaries of different classes produced by our DREAM are quite clear, demonstrating that our proposed DREAM is capable of producing high-quality and discriminative graph representations.

**Ablation Study.** We perform ablation studies to investigate the contributions of different components with six variants. In particular, we introduce six model variants: 1) DREAM w/o A: it replaces the attention mechanism with the global pooling. 2) DREAM w/o MM: it is trained without the multi-sample mixup framework. 3) DREAM w/o IB: it removes the interaction between two branches. 4) DREAM w/o OC: it ignores the open-set classifier during training. 5)

Table 3: Ablation study of different components in DREAM. $P1$ and $P2$ denote different domains in the datasets.

| Methods | Lette-high | | COIL-DEL | |
|---|---|---|---|---|
| | $P1{\to}P2$ | $P2{\to}P1$ | $P1{\to}P2$ | $P2{\to}P1$ |
| DREAM w/o A | 55.4 | 51.0 | 40.3 | 38.7 |
| DREAM w/o MM | 56.1 | 51.3 | 41.0 | 39.1 |
| DREAM w/o IB | 57.4 | 52.1 | 43.7 | 38.3 |
| DREAM w/o OC | 55.3 | 51.0 | 41.7 | 37.4 |
| DREAM w EG | 57.4 | 52.4 | 42.8 | 39.5 |
| DREAM w ES | 58.1 | 52.7 | 43.5 | 40.0 |
| DREAM | **58.7** | **53.3** | **44.0** | **40.2** |

DREAM w EG: it ensembles two graph-level representation branches with different parameters. 6) DREAM w ES: it ensembles two subgraph-enhanced branches with different parameters.

Table 3 shows the results of different variants and we can have the following observations. *Firstly*, DREAM w/o A achieves worse performance than DREAM, validating the effectiveness of the attention mechanism for super-nodes. *Secondly*, DREAM w/o MM performs worse than our full model. This can be attributed to the fact that the multi-sample mixup framework maps each instance to cross-domain samples, which effectively aligns the domain features and leads to better performance. *Thirdly*, by ignoring the interaction between branches, the performance of DREAM w/o IB decreases compared with DREAM on Letter-high and COIL-DEL datasets. The potential reason is that with the mutual supervision between different branches, DREAM tends to learn consistent results for classification, thus increasing the reliability of the results. *Fourthly*, DREAM w/o OC

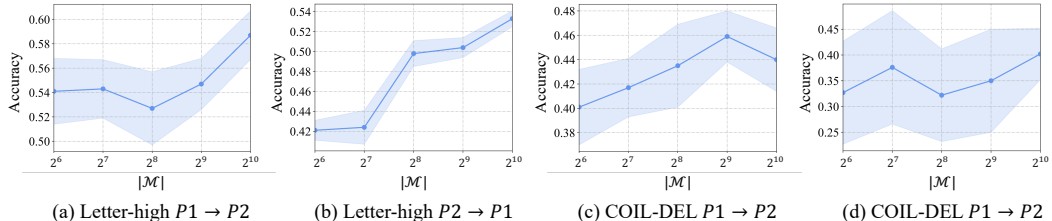

(a) Letter-high $P1 \to P2$  (b) Letter-high $P2 \to P1$  (c) COIL-DEL $P1 \to P2$  (d) COIL-DEL $P1 \to P2$

Figure 3: Hyperparameter sensitivity analysis of the memory bank capability $|\mathcal{M}|$ on Letter-high and COIL-DEL datasets. $|\mathcal{M}|$ is the memory capability and ranges from 64 to 1024. The solid line corresponds to the mean results and the shaded area represents the standard deviation.

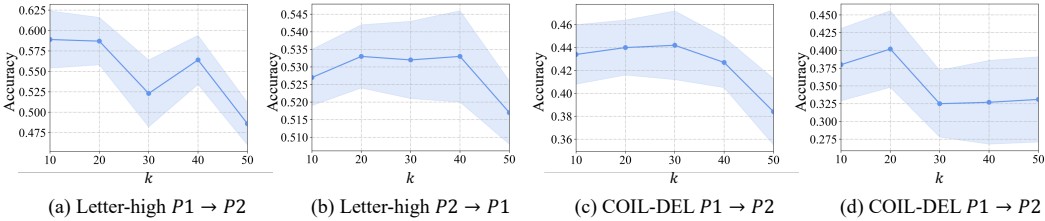

(a) Letter-high $P1 \to P2$  (b) Letter-high $P2 \to P1$  (c) COIL-DEL $P1 \to P2$  (d) COIL-DEL $P1 \to P2$

Figure 4: Hyperparameter sensitivity analysis of the similar samples $k$ on Letter-high and COIL-DEL datasets. $k$ denotes the number of similar samples sampled from the other domain. The solid line corresponds to the mean results and the shaded area represents the standard deviation.

removes the open-set classifier, which achieves the worst results mostly. The potential reason is that the open-set classifier introduces the unseen classes with Mixup into the closed-set settings, making the model more generalized to the unseen classes in the other domain. *Fifthly*, DREAM w EG and DREAM w ES perform worse than the full model, indicating the importance of interaction between two different branches from two complementary views.

**Sensitivity Analysis.** In this part, we investigate how the hyperparameters influence the performance of DREAM. In our implementation, $|\mathcal{M}|$ controls the number of historical features of the memory bank, and $k$ controls the number of similar samples in another domain corresponding to each sample. We set $|\mathcal{M}|$ and $k$ in $\{64, 128, 256, 512, 1024\}$ and $\{10, 20, 30, 40, 50\}$ with other parameters fixed, and the results are presented in Figure 3 and Figure 4. From the results, we can find that when the value of $|\mathcal{M}|$ rises, the performance tends to improve gradually. The potential reason is that with the increase of memory capability, DREAM will remember more historical features for domain alignment. However, with the limitation of GPU memories, too large $|\mathcal{M}|$ will lead to memory overflow. Therefore, the default value of $|\mathcal{M}|$ is set to 1024 for these datasets. Moreover, the accuracy of DREAM tends to increase first and decrease gradually when $k$ rises. This is because fewer $k$ cannot provide sufficient information for the domain alignment. When with larger $k$, one sample would collect much more samples from the other domain, which may introduce more noisy instances, leading the worse performance. Therefore, we set the $k$ to 20 as the default.

## 5 CONCLUSION

This work addresses the challenging problem of open-set graph domain adaptation and presents a novel method named DREAM. Our DREAM incorporates two branches, a graph-level representation learning branch and a subgraph-enhanced branch, which are trained jointly to explore graph topological structures from different perspectives and ensure inter-module consistency. To handle open-set scenarios, we generate virtual unknown samples belonging to novel classes for additional supervision in the open-set scenarios. To further mitigate domain shift, we construct a $k$ nearest neighbor-based graph-of-graph to generate cross-domain counterparts using multi-sample mixup, which helps to improve cross-domain consistency. Extensive experiments on six datasets validate the superiority of our proposed DREAM over existing state-of-the-art approaches. In our future work, we would extend our DREAM to more realistic problems such as universal graph domain adaptation and test-time graph domain adaptation.

ACKNOWLEDGEMENT

This work is supported by STI 2030—Major Projects (No. 2021ZD0201405).

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

## A    PROOF OF LEMMA 3.1

**Lemma 3.1.** $\mathbb{E}_{G \in \mathcal{D}^t}[\log p_\theta(G)] \geq \mathbb{E}_{G \in \mathcal{D}^t, y \sim q_\phi(y|G)}[\log \frac{p_\theta(G,y)}{q_\phi(y|G)}]$ where the equality holds when $p_\theta(y|G) = q_\phi(y|G)$.

*Proof.* We have:

$$
\begin{aligned}
\mathbb{E}_{G \in \mathcal{D}^t}[\log p_\theta(G)] &= \mathbb{E}_{G \in \mathcal{D}^t}[\log \sum_y p_\theta(G,y)] \\
&= \mathbb{E}_{G \in \mathcal{D}^t}[\log \sum_y q_\phi(y|G) \frac{p_\theta(G,y)}{q_\phi(y|G)}] \\
&\geq \mathbb{E}_{G \in \mathcal{D}^t}[\log \sum_y q_\phi(y|G) \log \frac{p_\theta(G,y)}{q_\phi(y|G)}] \\
&= \mathbb{E}_{G \in \mathcal{D}^t, y \sim q_\phi(y|G)}[\log \frac{p_\theta(G,y)}{q_\phi(y|G)}]
\end{aligned}
\tag{15}
$$

The inequality comes from the Jensen's inequality and the equality holds when $q_\phi(y|G)/p_\theta(G,y)$ is a constant. Since $p_\theta(G,y) = p(G)p_\theta(y|G)$, we have $q_\phi(y|G)/p_\theta(y|G)$ is a constant. Therefore, the equality holds when $p_\theta(y|G) = q_\phi(y|G)$. $\qquad\square$

## B    COMPLEXITY ANALYSIS

Here, we analyze the computational complexity of the proposed DREAM. The computational complexity primarily relies on two different branches for graph representations. Given a graph $G$, $||A||_0$ is the number of nonzeros in $A$, $d$ denotes the feature dimension, $L$ denotes the number of layers, $T$ denotes the number of views, $R$ denotes the number of clusters. The graph-level representation learning branch takes $O(L||A||_0 d + L|V|d^2 + Td|V| + Td^2)$. The subgraph-view representation learning branch takes $O(L||A||_0 d + L|V|d^2 + R^2 d)$. In our case, $R^2 \ll ||A||_0$, $T \ll d$ and $T \ll |V|$. Therefore, the complexity of the proposed DREAM and GraphCL are both $O(L||A||_0 d + L|V|d^2)$ for each graph sample, which is linearly related to $||A||_0$ and $|V|$. We would further explore the complexity of our method in more complicated scenarios in our future works.

## C    ALGORITHM

---
**Algorithm 1** Learning Algorithm of DREAM

---
**Input:** Source data $\mathcal{D}^s$; Target data $\mathcal{D}^t$.
**Output**: Parameters $\theta$ and $\phi$ for two branches.

 1: Warm up to initialize $\theta$ and $\phi$.
 2: **while** not convergence **do**
 3:     Update the memory bank $\mathcal{M}$;
 4:     Annotate target graph samples using $p_\theta(y|G)$ and $q_\phi(y|G)$;
 5:     // E-step
 6:     Optimize model parameters $\phi$ with fixed $\theta$;
 7:     // M-step
 8:     Optimize model parameters $\theta$ with fixed $\phi$.
 9: **end while**

---

## D    DATASETS

### D.1    DATASET DESCRIPTION

We conduct extensive experiments on six public benchmark graph datasets. Four of them, i.e., MSRC_21, Letter-high, COIL-DEL, COIL-RAG are from TUDataset. The rest two are released by

Table 4: Statistics of the experimental datasets.

| Dataset | #Graph | #Classes | #Avg. Nodes | #Avg. Edges | #Known | #Unknown |
|---------|--------|----------|-------------|-------------|--------|----------|
| MSRC_21 | 563 | 20 | 77.52 | 198.32 | 16 | 4 |
| Letter-high | 2,250 | 15 | 4.67 | 4.50 | 12 | 3 |
| COIL-DEL | 3,900 | 100 | 21.54 | 54.24 | 80 | 20 |
| COIL-RAG | 3,900 | 100 | 3.01 | 3.02 | 80 | 20 |
| MNIST | 70,000 | 10 | 70.57 | 564.53 | 8 | 2 |
| CIFAR10 | 60,000 | 10 | 117.63 | 941.07 | 8 | 2 |

Benchmark Dataset, i.e., MNIST and CIFAR10. The dataset statistics can be found in Table 4, and their details are shown as follows:

- **MSRC_21.** The images in MSRC_21 (Neumann et al., 2016) are represented by the conditional Markov random field graph, with nodes representing each image's superpixels and edges as the connection if the superpixels are adjacent.

- **Letter-high.** Letter-high (Riesen et al., 2008) consists of 15 capital letters with straight lines (i.e., A, E, F, H, I, K, L, M, N, T, V, W, X, Y, Z). A prototype line drawing is built for each class, and then converted into graphs with edges denoting lines and nodes denoting ending points.

- **COIL-DEL.** COIL-DEL (Riesen et al., 2008) applies the Harris corner detection and Delaunay Triangulation to extract corner features from images. The graph is described with lines by edges and their ending points by nodes. Each node has a 2D attribute and edges are unlabeled.

- **COIL-RAG.** COIL-RAG (Riesen et al., 2008) begins with segmenting each picture into regions of homogeneous color, and then transforms into graphs with nodes representing regions and edges representing the adjacency of regions. Each node is labeled with attributes specifying the color histogram and each edge is associated with the length of the common border of their corresponding adjacent regions.

- **MNIST.** MNIST (Dwivedi et al., 2020) is converted from the original MNIST dataset by using super-pixels as nodes and utilizing kNN to characterize the relationships between super-pixels.

- **CIFAR10.** CIFAR10 (Dwivedi et al., 2020) is constructed similarly to the MNIST dataset, which is more challenging since the large graphs with detailed semantic information.

### D.2 DATA PROCESSING

In our implementation, we process the datasets with different methods in corresponding models. For both branches, we process the four TUDataset by adding the self-loop connection of each node, and we use the one-hot embeddings for node attributes if the node features are not available. As for the Benchmark Datasets, we add the position information into the node features.

## E IMPLEMENTATION DETAILS

### E.1 BASELINES

In this part, we introduce the details of the compared baselines as follows:

**Graph Classification Methods.** We compare DREAM with seven popular graph classification approaches, i.e., GMT (Baek et al., 2021), GraphMix (He et al., 2022), GCN (Kipf & Welling, 2017), GraphSage (Hamilton et al., 2017), GIN (Xu et al., 2019), SAG Pooling (Lee et al., 2019) and GraphCL (You et al., 2020). For GCN and GIN, we use the Open Graph Benchmark[1] to implement the model. For GMT, GraphMix, GraphSage, SAG Pooling and GraphCL, we use the source codes provided by the corresponding paper. As for these baseline methods, we vary the dropout rate in the range of {0.1,0.3,0.5,0.7} and then choose the best one. The hidden dimension in these methods is set to 256 for a fair comparison.

---

[1]https://github.com/snap-stanford/ogb

- **GMT.** GMT (Baek et al., 2021) incorporates multi-head attention into the graph pooling layer, which enables the model to effectively consider the structural dependencies in the graph.

- **GraphMix.** GraphMix (He et al., 2022) is the abbreviation of Graph MLP-Mixer, which is capable of establishing long-range connections within graphs. It first divides each graph into a range of subgraphs with overlap and then leverages the self-attention mechanism to generate graph representations with node and patch position embeddings.

- **GCN.** GCN (Kipf & Welling, 2017) is the most popular approach to encoding the graph structure. It aggregates the neighborhood information to update the node representations in an iterative manner.

- **GraphSage.** GraphSage (Hamilton et al., 2017) aims to tend the transductive node classification to the inductive setting. It samples nodes from the neighborhood for each node instead of using all neighborhood nodes to increase computational efficiency. Through this, it can capture both the topological structure and the distribution of neighboring node features.

- **GIN.** GIN (Xu et al., 2019) is a powerful graph neural network which follows the message passing mechanism. It simply modifies the network architecture, which enhances its expressive capability in exploring diverse graph structures. It has been proven that the architecture has the same expressivity as the Weisfeiler-Lehman graph isomorphism test.

- **SAG Pooling.** SAG Pooling (Lee et al., 2019) first adopts the self-attention mechanism to learn the semantics of both node features and graph topology, and then masks unimportant nodes to reduce the number of nodes in each graph sample. The graph pooling layer is combined with the convolution layer and readout layer in the whole backbone.

- **GraphCL.** GraphCL (You et al., 2020) GraphCL is a popular framework for learning effective graph representations in a self-supervised manner. It adopts four augmentation strategies including node dropping, edge dropping, attribute masking and subgraph to generate different views of graph samples and then maximizes the mutual information between different views of the same graph.

**Graph Domain Adaptation Methods.** We compare DREAM with two state-of-the-art graph domain adaptation methods, i.e., DEAL (Yin et al., 2022) and CoCo (Yin et al., 2023a).

- **DEAL.** DEAL (Yin et al., 2022) is a domain adaptation method for graph classification, which contains an adversarial perturbation module and a pseudo-label distilling module. It adopts adversarial learning to transfer semantics from the source domain to the target domain.

- **CoCo.** CoCo (Yin et al., 2023a) consists of a graph convolutional network branch as well as a hierarchical graph kernel branch to explore topological information. To address the gap between the source and the target domain, it utilizes a cross-domain contrastive learning objective, which follows the EM algorithm.

**Open-world Graph Learning Methods.** We compare our DREAM with three open-world graph learning methods, i.e., RIGNN (Luo et al., 2023), OpenWGL (Wu et al., 2020b) and Open-WRF (Hoffmann et al., 2023).

- **RIGNN.** RIGNN (Luo et al., 2023) takes a rationale view to detect components containing the most information related to the label space and classify unlabeled graphs into a known class or an unseen class.

- **OpenWGL.** OpenWGL (Wu et al., 2020b) is a framework that utilizes the variational framework to learn effective node representations in open-world settings. We adapt the framework to our graph-level framework.

- **OpenWRF.** OpenWRF (Hoffmann et al., 2023) introduces a framework which decreases the sensitivity to thresholds in OOD detection.

**Open-set Recognition Methods.** We also compare our DREAM with three state-of-the-art methods, i.e., CSSR (Huang et al., 2022a), DIAS (Moon et al., 2022) and OSR (Vaze et al., 2022).

- **CSSR.** CSSR (Huang et al., 2022a) leverages an autoencoder-based architecture to measure whether instances belong to known classes or not while preserving high classification accuracy on known classes. It learns a specific autoencoder for each known category, which can produce an individual manifold as well.

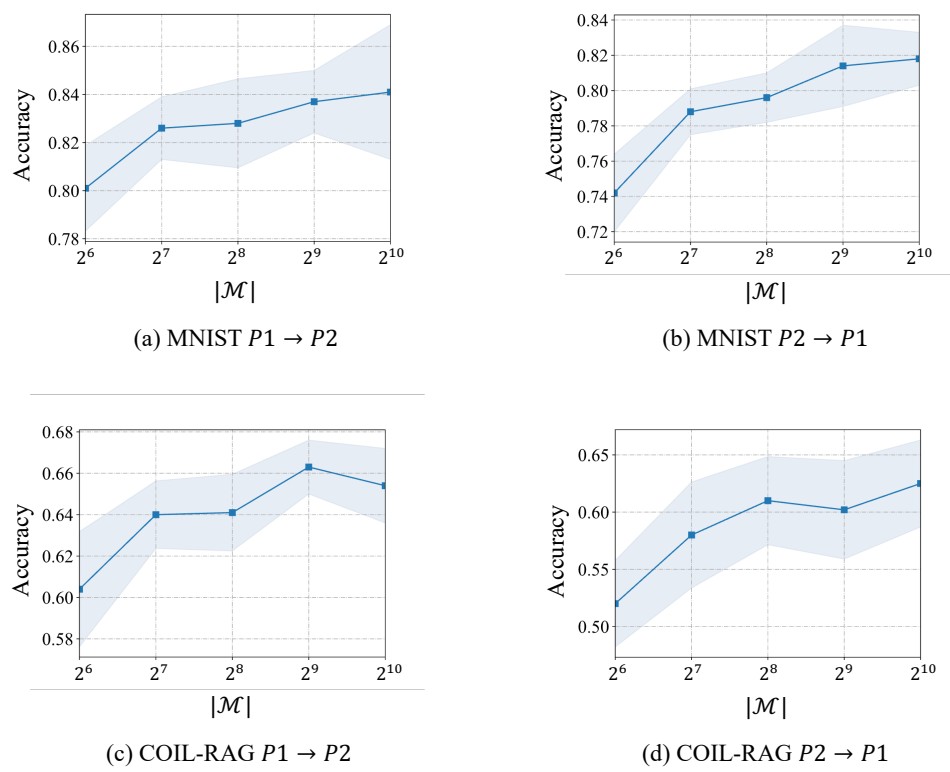

Figure 5: Hyperparameter sensitivity analysis of the memory bank capability $|\mathcal{M}|$ on MNIST and COIL-RAG datasets. $|\mathcal{M}|$ is the memory capability and ranges from 64 to 1024. The solid line corresponds to the mean results and the shaded area represents the standard deviation.

- **DIAS.** DIAS (Moon et al., 2022) aims to improve the capability of detecting unknown samples by introducing a difficulty-aware instance augmentation strategy. In particular, it improves the training procedure by generating both hard-difficulty and easy-difficulty examples to enhance the model's ability to handle unknown samples.

- **OSR.** OSR (Vaze et al., 2022) demonstrates that the performance of open-set recognition is highly related to close-set results. Then, it improves the baseline approaches using the maximum logit score to detect open-set samples, which can achieve state-of-the-art performance with limited modification.

### E.2   OUR IMPLEMENTATION

We implement our proposed DREAM with PyTorch (Paszke et al., 2017) and PyTorch Geometric library (Fey & Lenssen, 2019) on TUDataset[2] and Benchmark dataset[3]. We first pre-train the dual branches (i.e., graph-level and subgraph-enhanced branches) on the labeled domain, followed by the open-set classifier to train the network. Then, we start from the pre-trained model to implement the interaction between two branches and multi-sample mixup for domain alignment. Furthermore, to avoid memory overflow, we utilize the memory bank mechanism to store the previous embeddings and employ it to train DREAM on the unlabeled open-set dataset. Additionally, we search the number of sample neighbors from $\{10, 20, 30, 40, 50\}$, and the size of the memory bank from $\{64, 128, 256, 512, 1024\}$. During training, the hidden dimension of all the methods is set to 128, and we use the Adam (Kingma & Ba, 2014) optimizer and set the default batch size to 1024. The total number of training epochs is 200.

---

[2]https://chrsmrrs.github.io/datasets/

[3]https://data.pyg.org/datasets/benchmarking-gnns

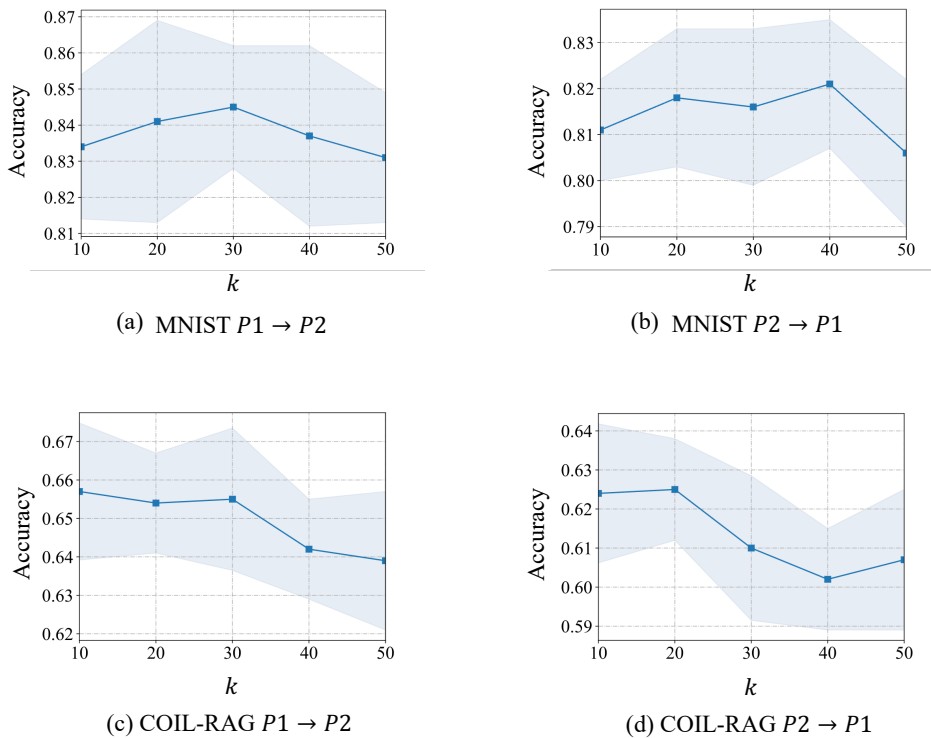

Figure 6: Hyperparameter sensitivity analysis of the similar samples $k$ on MNIST and COIL-RAG datasets. $k$ is the number of similar samples sampled from the other domain. The solid line corresponds to the mean results and the shaded area represents the standard deviation.

Table 5: Classification accuracy comparisons on four benchmark graph classification datasets (I). The best results are in boldface. $P1$ and $P2$ denote different domains in the datasets.

| Methods | MSRC_21 | | Letter-high | | COIL-DEL | | COIL-RAG | |
|---|---|---|---|---|---|---|---|---|
| | $P1{\to}P2$ | $P2{\to}P1$ | $P1{\to}P2$ | $P2{\to}P1$ | $P1{\to}P2$ | $P2{\to}P1$ | $P1{\to}P2$ | $P2{\to}P1$ |
| RIGNN | 72.6±4.3 | 73.8±2.4 | 51.1±3.2 | 43.8±2.1 | 34.2±2.9 | 33.5±3.3 | 57.6±2.1 | 57.2±3.1 |
| OpenWGL | 70.7±4.2 | 69.5±3.3 | 48.2±2.8 | 40.9±2.3 | 36.4±2.1 | 33.8±1.9 | 56.3±3.3 | 54.8±2.3 |
| OpenWRF | 73.0±3.7 | 70.4±2.5 | 42.6±2.6 | 40.4±1.9 | 31.5±1.8 | 30.2±2.8 | 55.3±2.4 | 54.8±2.4 |
| DEAL | 68.3±3.8 | 67.9±3.2 | 50.7±2.9 | 47.3±2.3 | 31.4±4.3 | 27.7±2.4 | 58.1±2.3 | 54.2±2.7 |
| CoCo | 69.8±4.7 | 68.3±2.1 | 50.3±3.4 | 49.8±1.4 | 33.7±3.1 | 19.8±1.7 | 61.3±1.4 | 55.7±1.6 |
| DREAM | **74.3±5.4** | **75.2±3.7** | **58.7±3.5** | **53.3±0.8** | **44.0±2.6** | **40.2±0.5** | **65.4±1.7** | **62.5±1.9** |

## F  MORE EXPERIMENTAL RESULTS

### F.1  MORE SENSITIVITY ANALYSIS

As introduced in Section 4.2, we set $|\mathcal{M}|$ and $k$ in {64, 128, 256, 512, 1024} and {10, 20, 30, 40, 50} with other parameters fixed, and conduct the experiments on MNIST and COIL-RAG. The results are presented in Figure 5 and Figure 6. We can have similar observations as in Letter-high and COIL-DEL in Section 4.2.

### F.2  MORE PERFORMANCE COMPARISON

In this part, we include five more additional baselines, i.e., RIGNN (Luo et al., 2023), OpenWGL (Wu et al., 2020b), OpenWRF (Hoffmann et al., 2023), DEAL (Yin et al., 2022) and CoCo (Yin et al., 2023a). From the results in Table 5, we can find that our proposed DREAM

Table 6: Ablation study of different components in DREAM. $P1$ and $P2$ denote different domains in the datasets. DREAM w/o MM removes the multi-sample mixup framework; DREAM w/o IB removes the interaction between two branches; DREAM w/o OC ignores the open-set classifier during training; DREAM w EG ensembles two graph-level representation branches with different parameters; DREAM w ES ensembles two subgraph-enhanced branches with different parameters.

| Methods | MSRC_21 | | COIL-RAG | | MNIST | | CIFAR10 | |
|---|---|---|---|---|---|---|---|---|
| | $P1{\rightarrow}P2$ | $P2{\rightarrow}P1$ | $P1{\rightarrow}P2$ | $P2{\rightarrow}P1$ | $P1{\rightarrow}P2$ | $P2{\rightarrow}P1$ | $P1{\rightarrow}P2$ | $P2{\rightarrow}P1$ |
| DREAM w/o MM | 72.4 | 72.8 | 62.2 | 60.3 | 82.4 | 79.6 | 45.6 | 44.5 |
| DREAM w/o IB | 73.2 | 74.1 | 64.7 | 61.1 | 82.3 | 80.4 | 47.1 | 45.3 |
| DREAM w/o OC | 70.3 | 70.7 | 61.3 | 59.1 | 80.8 | 77.4 | 44.2 | 42.3 |
| DREAM w EG | 73.1 | 73.7 | 64.4 | 61.5 | 82.7 | 80.4 | 47.0 | 45.4 |
| DREAM w ES | 73.8 | 74.4 | 64.7 | 61.3 | 82.4 | 80.7 | 47.1 | 46.8 |
| DREAM | **74.3** | **75.2** | **65.4** | **62.5** | **84.1** | **81.8** | **48.3** | **47.0** |

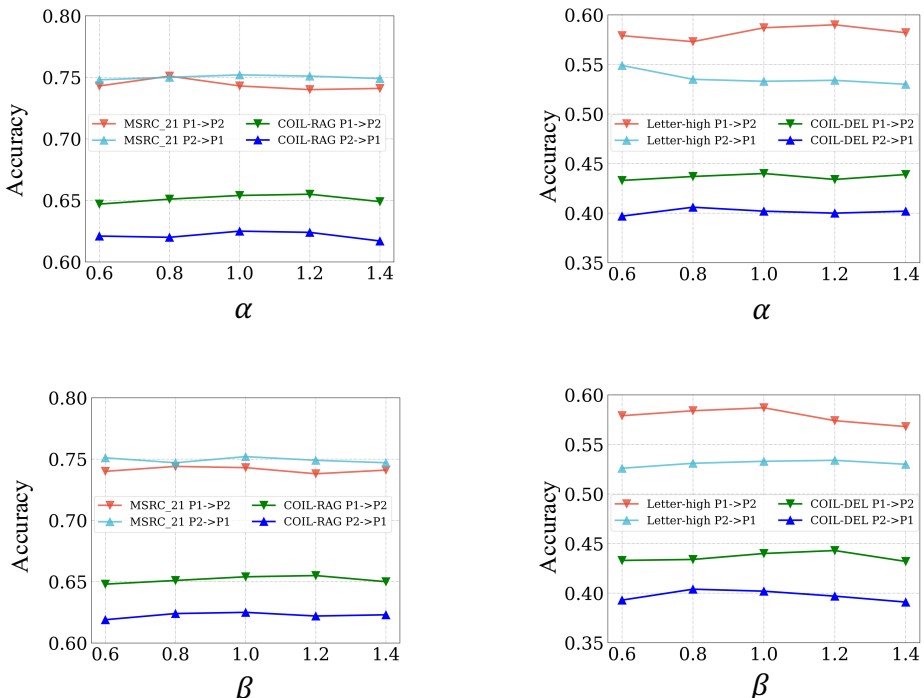

Figure 7: Hyperparameters analysis of $\alpha$ and $\beta$ on MSRC_21, COIL-RAG, Letter-high and COIL-DEL.

can achieve better performance in comparison to these baselines, which can validate the superiority of the proposed DREAM again.

### F.3 MORE ABLATION STUDIES

To validate the effectiveness of the different components, we conduct more experiments with five variants on different datasets, i.e., DREAM w/o MM, DREAM w/o IB, DREAM w/o OC, DREAM w EG and DREAM w. The results are shown in Table 6. From the results, we have similar observations as summarized in Section 4.2.

### F.4 HYPERPARAMETERS ANALYSIS

In this part, we provide additional parameter analysis for loss combination. Here, we add the weights by modifying the overall loss to $L = L_S + \alpha L_T + \beta L_D A$. To determine the hyper-parameters of $\alpha$

Table 7: The compared performance of RIGNN and DREAM on more complicated scenarios with diverse environments.

| Methods | MSRC_21 | | | | Letter-high | | | |
|---------|---------|---------|---------|---------|---------|---------|---------|---------|
| | $P1{\rightarrow}P2$ | $P1{\rightarrow}P3$ | $P1{\rightarrow}P4$ | $P1{\rightarrow}P5$ | $P1{\rightarrow}P2$ | $P1{\rightarrow}P3$ | $P1{\rightarrow}P4$ | $P1{\rightarrow}P5$ |
| RIGNN | 60.8 | 61.4 | 60.3 | 59.7 | 45.1 | 44.5 | 43.9 | 42.6 |
| DREAM | **63.2** | **63.8** | **61.1** | **62.1** | **47.3** | **45.9** | **46.8** | **44.6** |

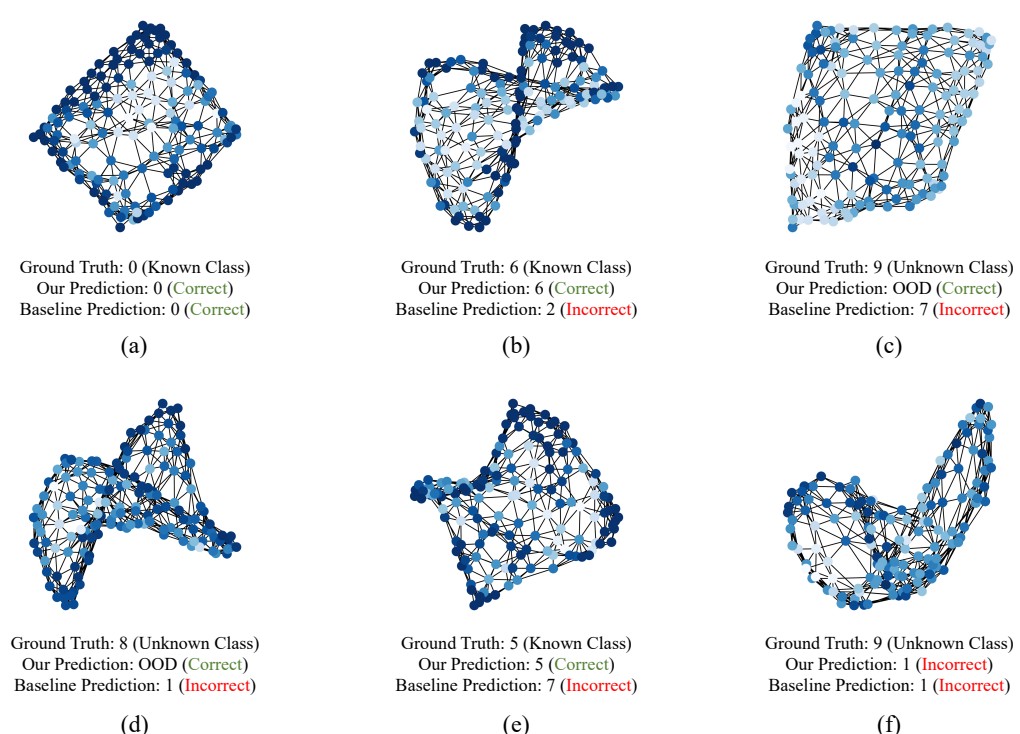

Figure 8: Visualization of graphs on the CIFAR10 dataset. We can observe that our model has the potential to detect unknown samples, which the baseline model cannot identify.

and $\beta$, we study the sensitivity analysis with $\alpha$ and $\beta$ in the range of $\{0.6, 0.8, 1.0, 1.2, 1.4\}$, which is shown in the Fig. 7. We can find that our performance is robust to both $\alpha$ and $\beta$. Therefore, we directly set the default parameter to $1.0$.

## F.5 COMPLICATED SCENARIO WITH DIVERSE ENVIROMENTS

To apply the proposed DREAM to more complicated scenarios, we separate the dataset into five parts, including four different target domains and one source domain for each dataset. The performance comparison of different methods is shown in Table 7. From the results, we can conclude that our DREAM achieves better generalization capacity in comparison with the baseline.

## F.6 VISUALIZATION OF GRAPH STRUCTURES

We provide a visualization of the graph structures in Figure 8 to further illustrate their complexity. We also provide the prediction results of our proposed DREAM and the baseline model GIN. It can be observed that the graphs possess intricate structures. It becomes apparent that employing only one branch, either the graph-level view or the subgraph view, is insufficient for effective representation learning. This observation supports the importance and necessity of utilizing both branches concurrently, which can enhance the quality of the learned representations. By leveraging

the complementary information from both views, we can capture a more comprehensive understanding of the underlying graph structures and improve the overall performance to detect samples from unknown classes. However, when graph structures are too complex (see Figure 8 (f)), both our proposed DREAM and the baseline would fail. Therefore, we will further improve our model to face more complicated scenarios.

