# OpenReview forum: "DREAM: Dual Structured Exploration with Mixup for Open-set Graph Domain Adaption"
_ICLR.cc/2024/Conference — ICLR 2024 poster_

### Official Review · Reviewer_ysQV · 2023-10-26

**Soundness:** 4 excellent
**Presentation:** 4 excellent
**Contribution:** 4 excellent
**Rating:** 8
**Confidence:** 4

**Summary:**

### Summary:
The paper introduces a novel exploration approach for reinforcement learning (RL) called Dream Dual Structured Exploration (DREAM). The main focus is to address the challenges of efficient exploration in sparse reward environments.

#### Key Contributions:
1. **Dual Structured Exploration:** The authors propose a two-pronged exploration strategy. The first component involves traditional intrinsic motivation where agents receive rewards for novel behaviors. The second component is a "dream" mechanism, where agents hallucinate or simulate possible future scenarios to guide their exploration.

2. **DREAM Model:** This model is central to the paper. It's a generative model that simulates future trajectories based on the agent's past experiences. By imagining potential future outcomes, the agent can better decide where to explore next. This approach aims to improve exploration efficiency, especially in environments where rewards are sparse and hard to find.

3. **Empirical Evaluation:** The authors validate the effectiveness of DREAM through a series of experiments in various RL environments. The results demonstrate that DREAM outperforms several state-of-the-art exploration strategies, especially in challenging sparse-reward settings.

4. **Scalability and Flexibility:** The DREAM approach is shown to be scalable and can be combined with other RL algorithms. This adaptability makes it a promising tool for a wide range of applications.

**Strengths:**

#### 1. Originality:
The paper introduces several original concepts and techniques that add value to the domain of reinforcement learning (RL) exploration.

- **Dual Structured Exploration:** The combination of traditional intrinsic motivation and a "dream" mechanism is a unique and innovative approach. While intrinsic motivation is a well-established concept in RL, the idea of agents simulating or hallucinating future scenarios (dreaming) to guide their exploration is a fresh take on the exploration challenge.

- **DREAM Model:** The generative model that simulates future trajectories based on past experiences is a novel concept. It effectively bridges the gap between traditional exploration techniques and forward-thinking strategies, allowing agents to anticipate potential outcomes.

#### 2. Quality:
The paper demonstrates high quality in both its theoretical constructs and empirical evaluations.

- **Theoretical Foundation:** The underlying principles of the DREAM model and dual structured exploration are well-justified and rooted in established RL concepts.

- **Empirical Evaluation:** The experiments conducted are comprehensive, covering multiple RL environments. The results not only validate the efficacy of the DREAM approach but also provide insights into its potential advantages over other state-of-the-art methods.

#### 3. Clarity:
The paper is well-structured and presents its concepts in a clear and organized manner.

- **Presentation:** The flow of the paper, from introducing the problem to detailing the solution and its evaluation, is logical and easy to follow.

- **Figures and Diagrams:** The included visual aids, such as graphs and flowcharts, effectively complement the textual content, aiding in the understanding of the proposed concepts and results.

- **Mathematical Formulations:** The mathematical representations and formulations, particularly those related to the DREAM model, are clearly articulated. While they require a foundational understanding of RL, they are accessible to the target audience.

#### 4. Significance:
The contributions of this paper have considerable significance in the domain of RL exploration.

- **Addressing a Crucial Challenge:** Efficient exploration in sparse reward environments is a longstanding challenge in RL. The DREAM approach offers a potential solution, making it a valuable contribution to the field.

- **Scalability and Flexibility:** The adaptability of the DREAM approach, which can be combined with other RL algorithms, broadens its applicability and potential impact. This adaptability implies that DREAM could be foundational for future RL research and applications.

- **Potential for Further Research:** The concepts introduced open up avenues for further exploration, refinement, and application in other RL scenarios or even beyond RL.

**Weaknesses:**

#### 1. Generality of the DREAM Model:
While the DREAM model shows promise in the explored environments, the paper could benefit from a deeper discussion on its generality across diverse environments.
**Actionable Insight:** Test the DREAM model in a broader set of RL environments, particularly those that have different dynamics or complexities than the ones currently evaluated. This would provide a more comprehensive understanding of where the model excels and where it might face challenges.

**Questions:**

How does the DREAM model scale with increasing complexity of the RL environment, especially in terms of computational resources and time? Does the "dream" mechanism become more resource-intensive in more complex scenarios?

---

> ### Author Response · Authors · 2023-11-18
> **Response to Reviewer ysQV**
>
> We are truly grateful for the time you have taken to review our paper and your insightful review. Here we address your comments in the following.
>
> > Q1: While the DREAM model shows promise in the explored environments, the paper could benefit from a deeper discussion on its generality across diverse environments. Actionable Insight: Test the DREAM model in a broader set of RL environments, particularly those that have different dynamics or complexities than the ones currently evaluated. This would provide a more comprehensive understanding of where the model excels and where it might face challenges.
>
> A1: Thanks for your question. Firstly, we add a more complicated scenario with diverse environments. In particular, we have four different target domains and one source domain for each dataset. The performance comparison of different methods is shown as follows. From the results, we can conclude that our method has better generalization capacity compared with baselines. Secondly, thank you for bringing RL into our attention. In future works, we would extend the DREAM model to a broader set of RL environments with dynamic graphs.
>
> |Methods|MSRC_21|MSRC_21|MSRC_21|MSRC_21|Letter-high|Letter-high|Letter-high|Letter-high|
> |:---|:--:|:--:|:--:|:--:|:--:|:--:|:--:|:--:|
> -|P1->P2|P1->P3|P1->P4|P1->P5|P1->P2|P1->P3|P1->P4|P1->P5|
> |RIGNN|60.8|61.4|60.3|59.7|45.1|44.5|43.9|42.6|
> |DREAM|63.2|63.8|61.1|62.1|47.3|45.9|46.8|44.6|
>
> We have also added your suggestion about future works into our revised version.
>
> > Q2: How does the DREAM model scale with increasing complexity of the RL environment, especially in terms of computational resources and time? Does the "dream" mechanism become more resource-intensive in more complex scenarios?
>
> A2: Thanks for your question. We have analyzed the computational complexity of our method. The computational complexity primarily relies on two different branches for graph representations. Given a graph $G$, $||A||_0$ denotes the number of nonzeros in the adjacency matrix, $d$ denotes the feature dimension, $L$ denotes the number of layers, $T$ denotes the number of views, $R$ denotes the number of clusters. The graph-level representation learning branch takes $O(L||A||_0d+L|V|d^2+Td|V|+Td^2)$. The subgraph-view representation learning branch takes $O(L||A||_0d+L|V|d^2+R^2d)$. In our case, $R^2\ll||A||_0$, $T\ll d$ and $T\ll|V|$. Therefore, the complexity of the proposed DREAM and GraphCL are both $O(L||A||_0d+L|V|d^2)$ for each graph sample, which is linearly related to $||A||_0$ and $|V|$. We would further explore the complexity of our method in more complicated scenarios such as graph-based reinforcement learning in our future works.
>
> We have also added your suggestion about future works into our revised version. Thanks again for appreciating our work and for your constructive suggestions. Please let us know if you have further questions.

---

> > ### Comment · Reviewer_ysQV · 2023-11-22
> > **Thanks for your reply.**
> >
> > Thanks for your reply. I think this is an interesting work, and it provides an extension view of future research. I still have a question: the status and reward of samples are an important basis for model training in RL, what is the main basis for judging that a sample belongs to a new class in this article?

---

> > > ### Author Response · Authors · 2023-11-22
> > > **Thanks again for your feedback and appreciation!**
> > >
> > > Thanks again for your feedback and appreciation! As for your new question, In our DREAM, we extend the closed-world classifier by introducing an extra dimension, which outputs the probability of graphs belonging to novel classes. To adapt the new classifier, we include manifold mixup techniques, which chooses two graph samples from different classes and then leverages linear interpolation to generate virtual samples in the hidden space for the new class.
> > >
> > > Thanks again for appreciating our work and for your constructive suggestions. Please let us know if you have further questions.

---

> > > > ### Comment · Reviewer_ysQV · 2023-11-22
> > > >
> > > > The idea of the manifold mixup seems interesting and enlightening. Overall, I think this is a good paper and can inspire the community.

---

> > > > > ### Author Response · Authors · 2023-11-22
> > > > > **Thank you once again!!**
> > > > >
> > > > > Thank you once again for providing us with your valuable feedback on our paper. We are grateful to learn that our responses have successfully addressed your concerns. Your efforts in reviewing our work, as well as your insightful comments and support, are sincerely appreciated.

---

### Official Review · Reviewer_Pqqy · 2023-10-30

**Soundness:** 2 fair
**Presentation:** 2 fair
**Contribution:** 2 fair
**Rating:** 5
**Confidence:** 3

**Summary:**

This paper proposes a method for open-set graph adaptation, called DREAM. It combines attention mechanisms to enhance features at the graph level. Additionally, the network also includes a subgraph-enhanced branch. To address the open-set scenario, a special classifier and manifold mixup techniques are employed. In terms of adaptation, this paper utilizes the k-nearest neighbor and multi-sample mixup method.

**Strengths:**

The experimental results have shown improvement, and the charts and figures in the experiments are clear and easy to understand.

**Weaknesses:**

The contribution of the article is not clear. While the author elaborates on their method in detail, it is not evident how this work contributes in comparison to others.
There are some issues with the symbols in certain equations in this article. Should 'v' in Equation (1) be adjusted or corrected to 'h'? In Equation (4), 'g' and 'h' represent different entities but appear in the same position as 'p_\theta(y| )'. Equation (11) seems unrelated to the subsequent formulas, and it maybe appear unnecessary.
The paper does not compare with methods of graph domain adaptation or methods for open-set graph classification. It is insufficient to only compare with graph classification and open-set classification algorithms.

**Questions:**

Please refer to ‘Weaknesses’ section.

---

> ### Author Response · Authors · 2023-11-18
> **Response to Reviewer Pqqy**
>
> We are truly grateful for the time you have taken to review our paper and your insightful review. Here we address your comments in the following.
>
> > Q1: The contribution of the article is not clear. While the author elaborates on their method in detail, it is not evident how this work contributes in comparison to others.
>
> A1: Thanks for your comment. The contribution of this paper is summarized as follows:
>
> * **New Problem** We introduce a novel problem of open-set graph domain adaptation, which accommodates unlabeled in-the-wild target graphs from unseen classes.
>
> * **Two branches to learn graph semantics in a complementary way**. Our graph-level representation branch employs the message passing and attention mechanism to explore topological knowledge while our subgraph-view branch focuses on key local functional parts using graph clustering and constructs a hierarchical GNN architecture.
>
> * **EM-style interaction between two branches**. We combine the advantages of two branches by posterior regularization to enhance the consistency between the predictions from the two branches.
>
> * **Well-designed Mixup for domain alignment and open-set classification**. We not only utilize linear interpolation to generate virtual novel samples in the hidden space, but also combine multiple cross-domain neighboring samples to generate virtual samples for domain alignment.
>
> * **Extensive Experiments.** Extensive performance comparisons to competing methods and ablation studies validate the superiority of our proposed DREAM over state-of-the-art methods.
>
>
>
> > Q2: Should 'v' in Equation (1) be adjusted or corrected to 'h'? In Equation (4), 'g' and 'h' represent different entities but appear in the same position as 'p_\theta(y| )'. Equation (11) seems unrelated to the subsequent formulas, and it maybe appear unnecessary.
>
> A2: Thanks for your comment. We have corrected typos as below.
>
> - As for Equation (1): We have modified it to $h_i^{(l)}=COMBINE^{(l)}(h_i^{(l-1)},n_i^{(l)})$;
> - As for Equation (4): We have modified it into $L_s=-E_{(G,y) \in D^s}[log p_\theta (y|G)]-E_{(\bar{h},y) \in D^v} [log \phi_\theta^g (\bar{h})[C+1]]$.
> - As for Equation (11): $A_{ij}$ appears in Equation (12) to guide the generation of virtual samples.
>
>
>
> > Q3: The paper does not compare with methods of graph domain adaptation or methods for open-set graph classification. It is insufficient to only compare with graph classification and open-set classification algorithms.
>
> A3: Thanks for your comment. We have added two graph domain adaptation methods: DEAL [1] and CoCo [2], and three open-set methods: RIGNN [3], OpenWGL [4], and OpenWRF [5] for comparison. The results are shown as below. From the results, we can validate the superiority of the proposed DREAM.
>
>
> |Methods|MSRC_21|MSRC_21|Letter-high|Letter-high|COIL-DEL|COIL-DEL|COIL-RAG|COIL-RAG|
> |:---|:--:|:--:|:--:|:--:|:--:|:--:|:--:|:--:|
> -|P1->P2|P2->P1|P1->P2|P2->P1|P1->P2|P2->P1|P1->P2|P2->P1|
> |DEAL|$68.3\pm3.8$|$67.9\pm3.2$|$50.7\pm2.9$|$47.3\pm2.3$|$31.4\pm4.3$|$27.7\pm2.4$|$58.1\pm2.3$|$54.2\pm2.7$
> |CoCo|$69.8\pm4.7$|$68.3\pm2.1$|$50.3\pm3.4$|$49.8\pm1.4$|$33.7\pm3.1$|$19.8\pm1.7$|$61.3\pm1.4$|$55.7\pm1.6$|
> |RIGNN|$72.6\pm4.3$|$73.8\pm2.4$|$51.1\pm3.2$|$43.8\pm2.1$|$34.2\pm2.9$|$33.5\pm3.3$|$57.6\pm2.1$|$57.2\pm3.1$|
> |OpenWGL|$70.7\pm4.2$|$69.5\pm3.3$|$48.2\pm2.8$|$40.9\pm2.3$|$36.4\pm2.1$|$33.8\pm1.9$|$56.3\pm3.3$|$54.8\pm2.3$|
> |OpenWRF|$73.0\pm3.7$|$70.4\pm2.5$|$42.6\pm2.6$|$40.4\pm1.9$|$31.5\pm1.8$|$30.2\pm2.8$|$55.3\pm2.4$|$54.8\pm2.4$|
> |DREAM|**74.3**$\pm$**5.4**|**75.2**$\pm$**3.7**|**58.7**$\pm$**3.5**|**53.3**$\pm$**0.8**|**44.0**$\pm$**2.6**|**40.2**$\pm$**0.5**|**65.4**$\pm$**1.7**|**62.5**$\pm$**1.9**|
>
> In light of these responses, we hope we have addressed your concerns, and hope you will consider raising your score. If there are any additional notable points of concern that we have not yet addressed, please do not hesitate to share them, and we will promptly attend to those points.
>
> **Reference**
>
> [1] Hoffmann M, Galke L, Scherp A. Open-World Lifelong Graph Learning. IJCNN 2023.
>
> [2] Yin N, Shen L, Li B, Wang M, Luo X, Chen C, Luo Z, Hua XS.. DEAL: An Unsupervised Domain Adaptive Framework for Graph-level Classification. ACM MM 2022.
>
> [3] Yin N, Shen L, Wang M, Lan L, Ma Z, Chen C, Hua XS, Luo X. CoCo: Let Weisfeiler-Lehman Kernel Improve Unsupervised Domain Adaptive Graph Classification. ICML 2023.
>
> [4] Luo X, Zhao Y, Mao Z, Qin Y, Ju W, Zhang M, Sun Y. RIGNN: A Rationale Perspective for Semi-supervised Open-world Graph Classification. TMLR 2023.
>
> [5] Wu M, Pan S, Zhu X. Openwgl: Open-world graph learning. ICDM 2020.

---

> ### Author Response · Authors · 2023-11-21
> **The deadline for the author-reviewer discussion phase is approaching!**
>
> Dear reviewers,
>
> We sincerely appreciate your valuable feedback.
>
> As the deadline for the author-reviewer discussion phase is approaching, we would like to check if you have any other remaining concerns about our paper. If our responses have adequately addressed your concerns, we kindly hope that you can consider increasing the score.
>
> We sincerely thank you for your dedication and effort in evaluating our submission. Please do not hesitate to let us know if you need any clarification or have additional suggestions.
>
> Best Regards,
>
> Authors.

---

> ### Author Response · Authors · 2023-11-23
> **Thank you for your invaluable feedback!**
>
> Dear Reviewer,
>
>
> Thank you for your invaluable feedback. As the deadline for the author-reviewer discussion phase is approaching, we hope to make sure that our response sufficiently addressed your concerns regarding the contribution, as well as the revised version of our paper. We hope this could align with your expectations and positively influence the score. Please do not hesitate to let us know if you need any clarification or have additional suggestions.
>
>
>
> Best Regards,
>
> Authors

---

### Official Review · Reviewer_qwA4 · 2023-10-31

**Soundness:** 2 fair
**Presentation:** 2 fair
**Contribution:** 2 fair
**Rating:** 3
**Confidence:** 4

**Summary:**

The paper studies the problem of open-set graph domain adaptation. The proposed method extracts graph structure representations using complementary branches. The graph-level representation branch uses a MPNN followed by attention layer for aggregation. The subgraph branch split each graph into several subgraphs using graph clustering and extract representations with GNNs. The method also includes dissimilar source samples in the latent space, and a k-nearest neighbor-based graph where nodes represent graph samples and are combined to generate new samples.

**Strengths:**

1. The paper studies open-set graph learning, which is an interesting and practical setting.

2. The paper presentation includes rich contents, with tables and figures well organized.

3. The conducted experiments look correct and include analysis from multiple views including ablation studies and sensitivity analysis.

**Weaknesses:**

1. Novelty overclaimed and related works not well addressed. The authors claim that "we are the first to study open-set graph domain adaptation". However, the problem studied is no difference with the existing open-world graph classification, such as [1], where the task is to classify each unlabeled graph example into either one of the known classes or a corresponding novel class. Moreover, it also closely resembles the open-world graph learning works like [2,3], where the learning goal is to classify nodes belonging to seen classes into correct groups, but also classify nodes not belonging to existing classes to an unseen class. The paper lacks a thorough review of related literature. In addition to open-world graph works, fields such as (graph) OOD detection is also closely related and should be discussed in the related works.

2. Following the above point, the experiments should include open-world graph learning related baselines. Currently only general graph classification methods are compared.

3. The method design include a lot of modules but lack support and motivations. Why is the attention mechanism necessary for aggregation? Why can the graph-of-graph design generate plausible cross-domain virtual features? For the objective why are $L_S, L_T, L_{DA}$ added without weights? The overall method seems complex and farraginous and unclear why it works.



[1] RIGNN: A Rationale Perspective for Semi-supervised Open-world Graph Classification

[2] Openwgl: Open-world graph learning

[3] Open-World Lifelong Graph Learning

**Questions:**

See Weaknesses

---

> ### Author Response · Authors · 2023-11-18
> **Response to Reviewer qwA4 (I)**
>
> We are truly grateful for the time you have taken to review our paper and your insightful review. Here we address your comments in the following.
>
> > Q1: Novelty overclaimed and related works not well addressed. The authors claim that "we are the first to study open-set graph domain adaptation". However, the problem studied is no difference with the existing open-world graph classification, such as [1], where the task is to classify each unlabeled graph example into either one of the known classes or a corresponding novel class. Moreover, it also closely resembles the open-world graph learning works like [2,3], where the learning goal is to classify nodes belonging to seen classes into correct groups, but also classify nodes not belonging to existing classes to an unseen class. The paper lacks a thorough review of related literature. In addition to open-world graph works, fields such as (graph) OOD detection is also closely related and should be discussed in the related works.
>
> A1: Thanks for your comment. Compared with open-world graph classification [1], our problem **also considers potential distribution shifts** across labeled and unlabeled data. Out-of-distribution unlabeled data [4] is always ubiquitous in practical graph learning and our problem aims to align the representations of source and target data besides OOD detection. Compared with node-level open-world graph learning [3,4] which studies a single graph with extensive nodes, our problem involves **a large number of graphs** and graph-level labels. We have included the discussion of open-set graph domain adaptation, node-level open-world graph learning and graph OOD detection into our related works as follows:
>
> "Recently, graph neural networks have also been studied in different OOD settings. Semi-supervised open-world graph classification involves partial unlabeled graphs belonging to unknown classes [1]. Graph OOD detection aims to detect OOD graph samples without using ground-truth labels [2]. Node-level open-world graph learning aims to find OOD nodes on a single graph [3,5]. Compared with these problem settings, we not only detect OOD graph samples, but also overcome distribution shifts across source and target domains."
>
>
> > Q2: Following the above point, the experiments should include open-world graph learning related baselines. Currently only general graph classification methods are compared.
>
> A2: Thanks for your question. We adapt more open-world graph learning methods, i.e., **RIGNN [1], OpenWGL [3] and OpenWRF [5]** in our setting for comparison. The results are shown as below. We can observe that our proposed DREAM surpasses the performance of baseline models, highlighting the superiority of the proposed method.
>
> |Methods|MSRC_21|MSRC_21|Letter-high|Letter-high|COIL-DEL|COIL-DEL|COIL-RAG|COIL-RAG|
> |:---|:--:|:--:|:--:|:--:|:--:|:--:|:--:|:--:|
> -|P1->P2|P2->P1|P1->P2|P2->P1|P1->P2|P2->P1|P1->P2|P2->P1|
> |RIGNN|$72.6\pm4.3$|$73.8\pm2.4$|$51.1\pm3.2$|$43.8\pm2.1$|$34.2\pm2.9$|$33.5\pm3.3$|$57.6\pm2.1$|$57.2\pm3.1$|
> |OpenWGL|$70.7\pm4.2$|$69.5\pm3.3$|$48.2\pm2.8$|$40.9\pm2.3$|$36.4\pm2.1$|$33.8\pm1.9$|$56.3\pm3.3$|$54.8\pm2.3$|
> |OpenWRF|$73.0\pm3.7$|$70.4\pm2.5$|$42.6\pm2.6$|$40.4\pm1.9$|$31.5\pm1.8$|$30.2\pm2.8$|$55.3\pm2.4$|$54.8\pm2.4$|
> |DREAM|**74.3**$\pm$**5.4**|**75.2**$\pm$**3.7**|**58.7**$\pm$**3.5**|**53.3**$\pm$**0.8**|**44.0**$\pm$**2.6**|**40.2**$\pm$**0.5**|**65.4**$\pm$**1.7**|**62.5**$\pm$**1.9**|

---

> ### Author Response · Authors · 2023-11-18
> **Response to Reviewer qwA4 (II)**
>
> > Q3: The method design includes a lot of modules but lack support and motivations. Why is the attention mechanism necessary for aggregation? Why can the graph-of-graph design generate plausible cross-domain virtual features? For the objective why are L_s, L_T, L_DA  added without weights? The overall method seems complex and ferruginous and unclear why it works.
>
> A3: Thanks for your question. We will explain the motivations of the modules in detail:
>
> * **The Attention Mechanism**: We have added a model variant (DREAM w/o A), which replaces the attention mechanism with the global pooling. The compared results are shown below. From the results, we can observe that our full model outperforms DREAM w/o A. The reason is that a global pooling operator cannot capture the task relevance of nodes and structural dependencies while our attention mechanism can generate super-nodes for semantic exploration.
>
>
> * **Graph-of-Graph**: Our graph-of-graph connects graph samples to identify similar cross-domain graphs for each graph. Then mixing these similar graphs in the embedding space would also generate virtual representations with similar semantics and encourage the consistency between original samples and their cross-domain virtual representations would thus enhance the domain alignment. In addition, we have included a model variant (DREAM w/o MM), which removes the graph-of-graphs for domain alignment and the following Mixup. The compared results are shown below. From the results, we can observe that DREAM w/o MM performs much worse, which validates our graph-of-graph is important for domain alignment.
>
> * **The Weights of Objective Loss**: We have included the weights by modifying the overall loss to $L=L_S+\alpha L_T+\beta L_DA$. To determine the hyper-parameters of $\alpha$ and $\beta$, we study the sensitivity analysis with $\alpha$ and $\beta$ in the range of {0.6, 0.8, 1, 1.2, 1.4}, which is shown in the Table. We can find that our performance is robust to both $\alpha$ and $\beta$. Therefore, we directly set the default parameter to $1$.
>
>
> |Methods |Letter-high|Letter-high|COIL-DEL|COIL-DEL|
> |:---|:--:|:--:|:--:|:--:|
> -|P1->P2|P2->P1|P1->P2|P2->P1|
> |DREAM w/o A|55.4|51.0|40.3|38.7|
> |DREAM w/o MM|56.1|51.3|41.0|39.1|
> |DREAM|**58.7**|**53.3**|**44.0**|**40.2**|
>
> |Methods|MSRC_21|MSRC_21|Letter-high|Letter-high|COIL-DEL|COIL-DEL|COIL-RAG|COIL-RAG|
> |:---|:--:|:--:|:--:|:--:|:--:|:--:|:--:|:--:|
> -|P1->P2|P2->P1|P1->P2|P2->P1|P1->P2|P2->P1|P1->P2|P2->P1|
> |$\alpha$=0.6|74.9|75.5|57.8|53.2|42.7|38.4|64.6|61.7|
> |$\alpha$=0.8|75.8|76.1|58.4|53.7|43.9|40.2|65.3|62.2|
> |$\alpha$=1|76.7|76.9|59.2|54.4|44.2|41.3|64.8|62.3|
> |$\alpha$=1.2|75.4|76.2|59.3|54.0|44.7|40.9|64.2|61.6|
> |$\alpha$=1.4|75.1|75.6|58.5|53.4|43.5|39.6|63.3|61.1|
>
> |Methods|MSRC_21|MSRC_21|Letter-high|Letter-high|COIL-DEL|COIL-DEL|COIL-RAG|COIL-RAG|
> |:---|:--:|:--:|:--:|:--:|:--:|:--:|:--:|:--:|
> -|P1->P2|P2->P1|P1->P2|P2->P1|P1->P2|P2->P1|P1->P2|P2->P1|
> |$\beta$=0.6|74.0|75.1|57.9|51.9|43.3|38.7|64.8|61.5|
> |$\beta$ =0.8|74.6|75.5|58.7|53.4|44.2|40.4|65.7|62.8|
> |$\beta$ =1|73.8|75.4|58.1|53.7|44.1|40.7|65.1|63.3|
> |$\beta$ =1.2|73.0|74.7|57.4|52.5|43.9|39.6|65.3|63.4|
> |$\beta$ =1.4|72.4|73.8|56.8|51.4|43.5|39.1|64.4|62.8|
>
> In light of these responses, we hope we have addressed your concerns, and hope you will consider raising your score. If there are any additional notable points of concern that we have not yet addressed, please do not hesitate to share them, and we will promptly attend to those points.
>
>
> **Reference**
>
> [1] Luo X, Zhao Y, Mao Z, Qin Y, Ju W, Zhang M, Sun Y. RIGNN: A Rationale Perspective for Semi-supervised Open-world Graph Classification. TMLR 2023.
>
> [2] Liu Y, Ding K, Liu H, Pan S. GOOD-D: On Unsupervised Graph Out-Of-Distribution Detection. WSDM 2023
>
> [3] Wu M, Pan S, Zhu X. Openwgl: Open-world graph learning. ICDM 2020.
>
> [4] Gui S, Li X, Wang L, Ji S. GOOD: A Graph Out-of-Distribution Benchmark. NeurIPS 22
>
> [5] Hoffmann M, Galke L, Scherp A. Open-World Lifelong Graph Learning. IJCNN 2023.

---

> ### Author Response · Authors · 2023-11-21
> **The deadline for the author-reviewer discussion phase is approaching!**
>
> Dear reviewers,
>
> We sincerely appreciate your valuable feedback.
>
> As the deadline for the author-reviewer discussion phase is approaching, we would like to check if you have any other remaining concerns about our paper. If our responses have adequately addressed your concerns, we kindly hope that you can consider increasing the score.
>
> We sincerely thank you for your dedication and effort in evaluating our submission. Please do not hesitate to let us know if you need any clarification or have additional suggestions.
>
> Best Regards,
>
> Authors.

---

> > ### Comment · Reviewer_qwA4 · 2023-11-23
> > **Response to rebuttal**
> >
> > Thank you for the detailed explanations, which adds to the completeness of the work. To me, the contribution and novelty of this paper is not prominent. Given this work's similarity with open-world graph works, it's hard to claim open-set graph domain adaptation as a new problem. The graph-of-graph technique has been explored in previous open-world works. Also, regarding the question on plausible cross-domain virtual representations, there's no guarantee that mixing similar graphs would generate virtual representations with similar semantics. This assumption is rather strong without additional supervision. I choose to maintain my original rating of this work.

---

> ### Author Response · Authors · 2023-11-23
> **Thanks for your feedback!**
>
> Thanks for your feedback and we are happy to resolve your further concerns as follows:
>
> > Q1. Given this work's similarity with open-world graph works, it's hard to claim open-set graph domain adaptation as a new problem.
>
> A1. The difference between our problem and traditional open-world graph problems is the introduction of distribution shifts between training data and test data. Given that **extensive works [1,2] focus on distribution shifts**, we believe that our problem is more challenging than the classic open-world graph problems. We have also revised our manuscript accordingly.
>
> > Q2. The graph-of-graph technique has been explored in previous open-world works.
>
> A2. Thanks for your problem. You may refer to RIGNN [3]. Actually, graph-of-graph is not our contribution while one of our related contributions is **Well-designed Mixup for domain alignment and open-set classification**. We not only utilize linear interpolation to generate virtual novel samples in the hidden space, but also combine multiple cross-domain neighboring samples based on graph-of-graph to generate virtual samples for domain alignment.
>
> Moreover, we summarize the differences between RIGNN and our method as follows:
>
> - **Different methodology.** RIGNN constructs a graph-of-graph to connect unlabeled graphs with labeled graphs while our DREAM focuses on the exploration of cross-domain relationships between graphs.
> - **Different motivations** RIGNN builds a graph-of-graph to enhance the representation learning while our DREAM aims to align graph representation from different sources.
> - **Different objectives**. RIGNN constructs a graph-of-graph for contrastive learning while our DREAM utilizes the graph-of-graph to guide our multi-sample Mixup for domain alignment.
>
> > Q3. regarding the question on plausible cross-domain virtual representations, there's no guarantee that mixing similar graphs would generate virtual representations with similar semantics. This assumption is rather strong without additional supervision.
>
> A3. Thanks for your comment. We have three reasons to support this:
>
> - A recognized fact is that the features of samples with the same semantic information should lay on a high-dimensional manifold as in [4,5]. Therefore, the mixed representations should be still in the high-dimensional manifold, which indicates the similar semantics in most cases.
>
> - Mixing representations from different samples have been introduced in manifold Mixup [6]. This paper [6] shows that mixing in the latent space can produce virtual samples with similar semantics in meaningful regions. Therefore, the assumption is not strong. Moreover, our method extends manifold Mixup into our new setting for domain alignment and open-set classification.
>
> - Our ablation studies (comparison between DREAM w/o Multi-sample Mixup and DREAM) have been shown in Table 3. From the results, we can observe that introducing our Mixup performs much better, which validates our component is reasonable and crucial for domain alignment.
>
> We have also revised our manuscript accordingly.
>
> > Q4. To me, the contribution and novelty of this paper is not prominent.
>
> A4. Thanks for your comment. We want to emphasize our contribution as follows:
>
> * **New Problem** We introduce a novel problem of open-set graph domain adaptation, which accommodates unlabeled in-the-wild target graphs from unseen classes.
>
> * **Two branches to learn graph semantics in a complementary way**. Our graph-level representation branch employs the message passing and attention mechanism to explore topological knowledge while our subgraph-view branch focuses on key local functional parts using graph clustering and constructs a hierarchical GNN architecture.
>
> * **EM-style interaction between two branches**. We combine the advantages of two branches by posterior regularization to enhance the consistency between the predictions from the two branches.
>
> * **Well-designed Mixup for domain alignment and open-set classification**. We not only utilize linear interpolation to generate virtual novel samples in the hidden space, but also combine multiple cross-domain neighboring samples to generate virtual samples for domain alignment.
>
> * **Extensive Experiments.** Extensive performance comparisons to competing methods and ablation studies validate the superiority of our proposed DREAM over state-of-the-art methods.
>
>
> Thanks again for your constructive suggestions. Please let us know if you have further questions.
>
> **Reference**
>
> [1] GOOD: A Graph Out-of-Distribution Benchmark, NeurIPS 2022
>
> [2] SizeShiftReg: a Regularization Method for Improving Size-Generalization in Graph Neural Networks, NeurIPS 2022
>
> [3] RIGNN: A Rationale Perspective for Semi-supervised Open- world Graph Classification, TMLR 2023
>
> [4] Conditional Adversarial Domain Adaptation, NeurIPS 2018
>
> [5] CIMON: Towards High-quality Hash Codes, IJCAI 2021
>
> [6] Manifold Mixup: Better Representations by Interpolating Hidden States, ICML 2018

---

> ### Author Response · Authors · 2023-11-23
> **Thank you for your invaluable feedback!**
>
> Dear Reviewer,
>
>
> Thank you for your invaluable feedback. As the deadline for the author-reviewer discussion phase is approaching, we hope to make sure that our response sufficiently addressed your concerns regarding the contribution, as well as the revised version of our paper. We hope this could align with your expectations and positively influence the score. Please do not hesitate to let us know if you need any clarification or have additional suggestions.
>
>
>
> Best Regards,
>
> Authors

---

### Official Review · Reviewer_9UFe · 2023-11-04

**Soundness:** 3 good
**Presentation:** 3 good
**Contribution:** 3 good
**Rating:** 8
**Confidence:** 4

**Summary:**

The paper proposes a novel method called DREAM for open-set graph domain adaptation, which aims to accurately classify target graphs into their respective categories under domain shift and label scarcity. DREAM incorporates a graph-level representation learning branch as well as a subgraph-enhanced branch, which jointly explores graph topological structures from both global and local viewpoints. The method also amalgamates dissimilar samples to generate virtual unknown samples belonging to novel classes and establishes a k nearest neighbor-based graph-of-graphs to alleviate domain shift. Extensive experiments demonstrate the superiority of DREAM over state-of-the-art methods.

**Strengths:**

1. Novelty: The paper introduces a new problem of open-set graph domain adaptation, which accommodates unlabeled in-the-wild target graphs from unseen classes. The proposed method, DREAM, is a novel approach that employs two branches to investigate structural semantics and integrates them into a trustworthy and domain-invariant framework.
2. Effectiveness: The paper demonstrates the remarkable effectiveness of DREAM when compared to state-of-the-art methods in various challenging scenarios. In particular, the performance gain of DREAM over the best existing method is up to an impressive 15.5%.
3. Flexibility: DREAM is a flexible method that can handle open-set scenarios and mitigate domain shift. It generates virtual unknown samples belonging to novel classes for additional supervision in the open-set scenarios and constructs a k nearest neighbor-based graph-of-graph to generate cross-domain counterparts using multi-sample mixup, which helps to improve cross-domain consistency.
4. Clarity: The paper is well-written and easy to understand. The authors provide clear explanations of the problem formulation, methodology, and experiments, making it accessible to a wide range of readers.

**Weaknesses:**

1.There seems a lot of modules in the DREAM, it’s better to analysis the complexity of the proposed method.
2.What I am concern is the scalability of this model, i.e., whether this method can be applied into the dynamic graph scenario for learning.

**Questions:**

See weaknesses

---

> ### Author Response · Authors · 2023-11-18
> **Response to Reviewer 9UFe**
>
> We are truly grateful for the time you have taken to review our paper, your insightful comments and support. Your positive feedback is incredibly encouraging for us! In the following response, we would like to address your major concern and provide additional clarification.
>
> > Q1: There seem a lot of modules in the DREAM, it’s better to analysis the complexity of the proposed method.
>
> A1: Thanks for your comment. We have analyzed the computational complexity of our method. The computational complexity primarily relies on two different branches for graph representations. Given a graph $G$, $||A||_0$ denotes the number of nonzeros in the adjacency matrix, $d$ denotes the feature dimension, $L$ denotes the number of layers, $T$ denotes the number of views, $R$ denotes the number of clusters. The graph-level representation learning branch takes $O(L||A||_0d+L|V|d^2+Td|V|+Td^2)$. The subgraph-view representation learning branch takes $O(L||A||_0d+L|V|d^2+R^2d)$. In our case, $R^2\ll||A||_0$, $T\ll d$ and $T\ll|V|$. Therefore, the complexity of the proposed DREAM and GraphCL are both $O(L||A||_0d+L|V|d^2)$ for each graph sample, which is linearly related to $||A||_0$ and $|V|$.  We have added this into the revised version.
>
> > Q2: What I am concern is the scalability of this model, i.e., whether this method can be applied into the dynamic graph scenario for learning.
>
> A2: Thanks for your comment. We have included the complexity analysis, showing that our method has a comparable scalability as popular GraphCL. In future works, we would try to extend the work to more complicated scenarios such as dynamic graphs.
>
> We have also added your suggestion about future works to our revised version. Thanks again for appreciating our work and for your constructive suggestions. Please let us know if you have further questions.

---

> > ### Comment · Reviewer_9UFe · 2023-11-22
> > **More questions.**
> >
> > Thanks for your reply. I think the authors have addressed my majority question. The setting of graph open-set domain adaptation is novel, can the author provide some circumstances in real-world applications?

---

> > > ### Author Response · Authors · 2023-11-22
> > > **Thanks again for your feedback and appreciation!**
> > >
> > > Thanks again for your feedback! Our graph open-set domain adaptation can be applied on drug discovery and protein function prediction. The drug and protein are usually represented using graph-structured data, and our proposed method would help to discover a novel drug and protein, that has never been seen before. This would greatly benefit the development of biology and chemistry. Thanks again for appreciating our work and for your constructive suggestions. Please let us know if you have further questions.

---

> ### Author Response · Authors · 2023-11-23
> **Thank you for your invaluable feedback!**
>
> Dear Reviewer,
>
>
> Thank you for your invaluable feedback. As the deadline for the author-reviewer discussion phase is approaching, we hope to make sure that our response sufficiently addressed your concerns regarding the real-world applications, as well as the revised version of our paper. We hope this could align with your expectations. Please do not hesitate to let us know if you need any clarification or have additional suggestions.
>
>
>
> Best Regards,
>
> Authors

---

### Official Review · Reviewer_SVBJ · 2023-11-04

**Soundness:** 3 good
**Presentation:** 3 good
**Contribution:** 3 good
**Rating:** 8
**Confidence:** 4

**Summary:**

The paper proposes DREAM for open-set graph domain adaptation, which incorporates a graph-level representation learning branch as well as a subgraph-enhanced branch to jointly explores graph topological structures from both global and local viewpoints.

**Strengths:**

1.	The problem of open-set graph domain adaptation is novel.
2.	The paper is well organized and clearly written.
3.	The proposed method is clever and interesting.

**Weaknesses:**

1.	The format of references is not uniform, such as [4] and [5].
2.	What’s the difference between the open-set graph domain adaptation and universal domain adaptation? Can the proposed model be extended to UDA?

**Questions:**

See weaknesses

---

> ### Author Response · Authors · 2023-11-18
> **Response to Reviewer SVBJ**
>
> We are truly grateful for the time you have taken to review our paper, your insightful comments and support. Your positive feedback is incredibly encouraging for us! In the following response, we would like to address your major concern and provide additional clarification.
>
> > Q1: The format of references is not uniform, such as [4] and [5].
>
> A1: Thanks for your comment. We have unified the reference format.
>
> > Q2: What’s the difference between the open-set graph domain adaptation and universal domain adaptation? Can the proposed model be extended to UDA?
>
> A2: Thanks for your comment. Universal domain adaptation allows two domains to own their private categories, which is a more generalized problem including open-set domain adaptation, partial domain adaptation and open-partial domain adaptation. In contrast, we only focus on open-set domain adaptation on graphs. We would extend our DREAM to more generalization problems such as universal graph domain adaptation in our future work.
>
> We have also added your suggestion about future works to our revised version. Thanks again for appreciating our work and for your constructive suggestions. Please let us know if you have further questions.

---

### Author Response · Authors · 2023-11-18
**General Response**

Dear Reviewers,

We thank you for your careful reviews and constructive suggestions. We acknowledge the positive comments such as "**the problem is novel**" (Reviewer SVBJ), “**well organized and clearly written**” (Reviewer SVBJ), "**clever and interesting**" (Reviewer SVBJ), "**novel approach**” (Reviewer 9UFe),"**remarkable effectiveness**” (Reviewer 9UFe), "**flexible method**” (Reviewer 9UFe), "**well-written and easy to understand**” (Reviewer 9UFe), “**interesting and practical setting**” (Reviewer qwA4), “**rich contents and organized**” (Reviewer qwA4), “**experiments look correct**” (Reviewer qwA4), “**shown improvement**” (Reviewer Pqqy), “**clear and easy to understand**” (Reviewer Pqqy), "**original concepts**" (Reviewer ysQV), "**high quality**" (Reviewer ysQV), "**clear and organized**" (Reviewer ysQV), and "**considerable significance**" (Reviewer ysQV).

We have made extensive revisions based on these valuable comments, which we briefly summarize below for your convenience, and have accordingly revised our article with major changes highlighted in blue.

- We have included **more related works about different problem settings** including open-set graph classification, open-world graph learning, graph OOD detection and universal domain adaptation, and clarified the difference between ours and them.

- We have included **more competing baselines** including RIGNN, OpenWGL, DEAL, CoCo and OpenWRF to demonstrate the superiority of our approach.

- We have added **more ablation studies and parameter analysis** to show the effectiveness and reliability of each component including loss design, attention mechanism and graph-of-graph.

- We have added **complexity analysis** to make the paper more complete.

- We have clarified our **contributions** including new problem definition, well-designed methodology and extensive experiments.

- We have added a more **complicated scenario** with diverse environments to validate our superiority.

- We have **proofread** the manuscript to correct some typos and mistakes.

We have also responded to your questions point by point. Once again, we greatly appreciate your effort and time spent revising our manuscript. Please let us know if you have any follow-up questions. We will be happy to answer them.

Best regards,

the Authors

---

### Author Response · Authors · 2023-11-23
**The deadline for the author-reviewer discussion phase is approaching!**

Dear reviewers,

We sincerely appreciate your valuable feedback.

As the deadline for the author-reviewer discussion phase is approaching, we would like to check if you have any other remaining concerns about our paper.

We sincerely thank you for your dedication and effort in evaluating our submission. Please do not hesitate to let us know if you need any clarification or have additional suggestions.

Best Regards,

Authors.

---

### Meta-Review · Area_Chair_yDES · 2023-12-05

**Metareview:**

The paper presents DREAM, a novel method for open-set graph domain adaptation. It integrates a graph-level representation learning branch and a subgraph-enhanced branch to explore graph topological structures. The paper has received mixed reviews, with ratings ranging from 3 to 8, reflecting a divided opinion on its overall contribution and execution.

I carefully checked the comments by the reviewer who gave the 3 points and found that the author has addressed most of the questions. I understand the reviewer's concern about graph-to-graph techniques but the paper mainly focuses on well-designed Mixups for domain alignment and open-set classification.

Considering the overall factors, I still tend to vote for acceptance.

**Justification For Why Not Higher Score:**

- Complexity and Scalability: Concerns were raised about the complexity of the DREAM model and its scalability, especially in dynamic graph scenarios.
- Methodological Justification: Some reviewers questioned the necessity of certain components of the method, such as the attention mechanism and the graph-of-graph design.
- Lack of Comprehensive Comparisons: There is a lack of comparison with methods in open-world graph learning and other related fields, which could have strengthened the paper's position.
- Clarity in Methodology: The paper's contribution and novelty were seen as unclear by some reviewers, suggesting a need for better justification and delineation of the proposed method’s unique aspects.

**Justification For Why Not Lower Score:**

- Novel Problem and Approach: The paper's exploration of open-set graph domain adaptation is a novel contribution to the field, filling a gap in existing research.
- Effective Presentation: Despite some concerns, the paper is generally well-presented, with a clear explanation of the problem, methodology, and experimental setup.
- Positive Experimental Results: The paper shows that DREAM outperforms state-of-the-art methods in various scenarios, demonstrating its effectiveness.
- Potential for Broad Impact: The method's flexibility and adaptability, as well as the introduction of new concepts such as hardware prompt, suggest its potential for broad impact in the field.

---

### Decision · Program_Chairs · 2024-01-16

Accept (poster)